# Global 3-D distribution of aerosol composition by synergistic use of CALIOP and MODIS observations

Rei Kudo[1], Akiko Higurashi[2], Eiji Oikawa[1], Masahiro Fujikawa[3], Hiroshi Ishimoto[1], and Tomoaki Nishizawa[2]

[1]Meteorological Research Institute, Japan Meteorological Agency, Tsukuba, 305-0052, Japan
[2]National Institute for Environmental Studies, Tsukuba, 305-8506, Japan
[3]Research Institute for Applied Mechanics, Kyusyu University, Kasuga, 816-8580, Japan

*Correspondence to*: Rei Kudo (reikudo@mri-jma.go.jp)

**Abstract.** For the observation of the global three-dimensional distribution of aerosol composition and the evaluation of shortwave direct radiative effect (SDRE) by aerosols, we developed a retrieval algorithm that uses observation data of the Cloud-Aerosol Lidar with Orthogonal Polarization (CALIOP) onboard the Cloud Aerosol Lidar Infrared Pathfinder Satellite Observations (CALIPSO) satellite, and the Moderate Resolution Imaging Spectroradiometer (MODIS) onboard Aqua. The CALIOP-MODIS retrieval optimizes the aerosol composition to both the CALIOP and MODIS observations in the daytime. Aerosols were assumed to be composed of four aerosol components: water-soluble (WS), light-absorbing (LA), dust (DS), and sea salt (SS) particles. The outputs of the CALIOP-MODIS retrieval are the vertical profiles of the extinction coefficient ($\alpha_a$), single-scattering albedo ($\omega_0$), and asymmetry factor ($g$) of total aerosols (WS+LA+DS+SS), and $\alpha_a$ of WS, LA, DS, and SS. Daytime observations of CALIOP and MODIS in 2010 were analysed by the CALIOP-MODIS retrieval. The global means of the aerosol optical depth ($\tau_a$) at 532 nm were $0.147 \pm 0.148$ for total aerosols, $0.072 \pm 0.085$ for WS, $0.027 \pm 0.035$ for LA, $0.025 \pm 0.054$ for DS, and $0.023 \pm 0.020$ for SS. $\tau_a$ of the CALIOP-MODIS retrieval was between those of the CALIPSO and MODIS standard products and was close to the MODIS standard product. The global means of $\omega_0$ and $g$ were $0.940 \pm 0.038$ and $0.718 \pm 0.037$; these values are in the range of those reported by previous studies. The horizontal distribution of each aerosol component was reasonable; for example, DS was large in desert regions, and LA was large in the major regions of biomass-burning and anthropogenic aerosol emissions. The values of $\tau_a$, $\omega_0$, $g$, and fine and coarse median radii of the CALIOP-MODIS retrieval were compared with those of the AERONET products. $\tau_a$ at 532 and 1064 nm of the CALIOP-MODIS retrieval agreed well with the AERONET products. The $\omega_0$, $g$, and fine and coarse median radii of the CALIOP-MODIS retrieval were not far from those of the AERONET products, but the variations were large, and the coefficients of determination for linear regression between them were small. In the retrieval results for 2010, the clear sky SDRE values for total aerosols at the top and bottom of the atmosphere were $-4.99 \pm 3.42$ and $-13.10 \pm 9.93$ W m$^{-2}$, respectively, and the impact of total aerosols on the heating rate was from 0.0 to 0.5 K day$^{-1}$. These results are generally similar to those of previous studies, but the SDRE at the bottom of the atmosphere is larger than that reported previously. Consequently, comparison with previous

studies showed that the CALIOP-MODIS retrieval results were reasonable with respect to aerosol composition, optical properties, and the SDRE.

## 1 Introduction

Aerosols have significant impacts on climate change through modification of the atmospheric radiation budget by scattering and absorbing solar and terrestrial radiation (aerosol-radiation interaction) and by modifying cloud physical properties (aerosol-cloud interaction). However, large uncertainties remain in evaluations of the aerosol impact on global warming (Arias et al., 2021) because of the large spatiotemporal variations in aerosol composition and the complex physical processes of aerosol-radiation and aerosol-cloud interactions. Because the radiative forcing of almost all aerosol chemical components is negative, aerosols contribute to the suppression of global warming; however, the radiative forcing of light-absorbing aerosols such as black carbon (BC) is positive (e.g., Matsui et al., 2018). Observations of spatiotemporal variations of aerosol composition are therefore essential for better understanding of the impacts of aerosols on climate change.

Based on the recent sophisticated numerical models with aerosol modules, and space- and ground-based observations, the data sets of aerosol composition climatology have been developed. The Modern-Era Retrospective analysis for Research and Applications version 2 (MERRA-2; Gelaro et al., 2019), and the Copernicus Atmosphere Monitoring Service Reanalysis (CAMSRA; Innes et al., 2019), and the Japanese Reanalysis for Aerosol v1.0 (JRAero; Yumimoto et al., 2017) are the reanalysis data sets by data assimilation schemes. The Max-Planck-Aerosol Climatology version 2 (MACv2; Kinne et al., 2019) is a climatology data set created by merging the data of the Aerosol Robotics Network (AERONET; Holben et al., 1998) and MAN (Smirnov et al., 2009) ground-based sun-photometer networks onto the ensemble mean of AeroCom models (Kinne et al., 2006). These data sets provide the global distributions of major aerosols, such as, sulfate, organic carbon, BC, dust, and sea-salt. The ModIs Dust AeroSol (MIDAS; Gkikas et al., 2021) data set is the global map of dust at fine resolution (0.1°×0.1°), and is created by the aerosol optical depth derived from the Moderate Resolution Imaging Spectroradiometer (MODIS) and the dust fraction of the MERRA-2 reanalysis. Amiridis et al. (2015) develop LIVA (Lidar climatology of Vertical Aerosol Structure for space-based lidar simulation studies), which is a three-dimensional multi-wavelength global aerosol and cloud optical data set. This data set is based on the Cloud-Aerosol Lidar with Orthogonal Polarization (CALIOP) on board the Cloud Aerosol Lidar Infrared Pathfinder Satellite Observations (CALIPSO) satellite (Winker et al., 2010), and the ground-based networks of European Aerosol Research Lidar Network (EARLINET; Bösenberg et al., 2003; Pappalardo et al., 2014) and AERONET.

These data sets are based on the combinations of numerical models with aerosol modules, and space- and ground-based remote sensing products. The remote sensing of aerosols plays an important role in constructing the data sets. Several ground-based remote sensing methods to retrieve aerosol composition have been developed. Kudo et al. (2010a) estimated 10-year variations of water-soluble particles (WS), BC, dust (DS), and sea salt (SS) from the direct and diffuse solar radiation in the visible and near infrared wavelength regions measured by two pyranometers and two pyrheliometers. Nishizawa et al.

(2007, 2008, 2011, 2017) retrieved concentrations of WS, BC, DS, and SS by using conventional Mie-scattering lidar as well as high-spectral-resolution lidar or Raman lidar data from the Asian Dust and Aerosol Lidar Observation Network (AD-Net; Sugimoto et al., 2015; Shimizu, et al., 2016). AERONET is an observational network of sun-sky radiometers that provides aerosol optical depth ($\tau_a$), single-scattering albedo ($\omega_0$), asymmetry factor ($g$), phase function, and complex refractive index data products (Dubovik and King, 2000; Dubovik et al., 2006; Synuk, et al., 2020). Schuster et al. (2005) and Dey et al. (2006) inferred BC concentrations from the AERONET-retrieved size distribution and complex refractive index. They considered internal and external mixtures of BC, sulfate, organic carbon, DS, and water. Satellite remote sensing has also been used for estimating aerosol composition and investigating global distributions. For example, Higurashi and Nakajima (2002) and Kim et al. (2007) retrieved the spatiotemporal distributions of sulfate, carbonaceous, DS, and SS aerosols from spectral information on radiances observed by satellite imagers, such as Sea-Viewing Wide Field-of-View Sensor (SeaWIFS), MODIS, and Ozone Monitoring Instrument (OMI). The CALIOP onboard the CALIPSO satellite has been utilized to classify aerosols at different altitudes (Omar et al., 2009; Winker et al., 2010). CALIOP Version 4 products classify eleven aerosol types: clean marine, DS, polluted continental/smoke, clean continental, polluted DS, elevated smoke, and dusty marine for tropospheric aerosols, and polar stratospheric aerosol, volcanic ash, sulfate/other, and smoke for stratospheric aerosols (Kim et al., 2018). These ground- and space-based methods assume that aerosols consist of a few components with different sizes, light-absorbing features, and shapes (spherical or non-spherical), and they retrieve the aerosol composition from optical measurements made by using different wavelengths and polarization.

The above-mentioned remote sensing methods retrieve aerosol data obtained by a single instrument. Recently, synergistic remote sensing methods using active and passive sensors have been developed. Passive sensors such as spectral radiometers and polarimeters provide the columnar properties of aerosols, whereas aerosol vertical profiles are obtained by active sensing by lidar. The LIRIC (Chaikovsky et al., 2016) and GARRLiC (Lopatin et al., 2013) algorithms retrieve the vertical profiles of aerosol physical and optical properties from lidar and AERONET sun-sky radiometer observations. SKYLIDAR (Kudo et al., 2016) estimates aerosol vertical profiles from AD-Net lidar and SKYNET sky radiometer observations (Nakajima et al., 2020). Xu et al. (2021) have retrieved aerosol physical and optical properties and ocean parameters such as chlorophyll $a$ concentration and surface wind speed from lidar and polarimetric observations over the ocean obtained during the ORACLES field campaign (Redemann et al., 2021).

To observe the global three-dimensional distribution of the aerosol composition, we have developed a new aerosol composition retrieval method that use the CALIOP and MODIS observations. The CALIOP-MODIS retrieval optimizes the aerosol composition to both the CALIOP and MODIS observations in the daytime. The columnar properties of aerosols are available from the MODIS multi-wavelength information, and $\tau_a$ is retrieved accurately (e.g., Shi et al., 2019), but aerosol vertical profiles cannot be obtained, and strong surface reflection (e.g., snow, desert) makes the retrieval difficult (Hsu et al., 2013). CALIOP observations exclude the data at the layers contaminated by the surface reflection and provide information on the vertical profiles of aerosol optical properties and particle shapes (spherical/non-spherical), but only limited wavelength information. Additionally, CALIOP does not detect the tenuous layers in the daytime due to the low signal to noise ratio. This

results in the underestimation of $\tau_a$ (Omar et al., 2013; Kim et al., 2018). The synergistic use of both instruments decreases the influences of the surface reflection and provide the more accurate columnar properties and vertical profiles of aerosols. Furthermore, the particle size information is obtained from the combined spectral information of the CALIOP and MODIS

observations (Kaufman et al., 2003).

In the previous remote sensing methods of aerosol compositions, there are two approaches in assuming aerosol components. One is the CALIOP-type categorization, such as, clean marine, polluted continental, and smoke, etc. These types are based on the aerosol characteristics observed in the typical scenes. The other is the similar categorization to the numerical models, i.e., sulfate, organic carbon, BC, DS, and SS. We adopted the latter approach because the external mixing of these

components is applicable to various scenes, and the $\tau_a$ and extinction coefficient ($\alpha_a$) of each component are suited for the comparison with the numerical models and the data assimilation. In this study, aerosols are assumed to consist of four components with different sizes, light-absorbing features, particle mixtures, and shapes. We defined these components as WS, light-absorbing particles (LA), DS, and SS. WS is defined by an external mixture of sulfate, and organic carbon, etc., because both the sulfate and organic carbon are fine and less light-absorbing particles, and it is difficult to estimate sulfate and organic

carbon separately from the MODIS and CALIOP measurements. LA is defined by an internal mixture of WS and BC. The details of the assumed aerosols are described in the Sect. 3. In this study, the global three-dimensional distributions of these components were estimated from the CALIOP-MODIS retrieval.

The aerosol-induced effects on the radiation field are denoted as aerosol radiative effects and are evaluated by the anomalies with respect to a reference state (Korras-Carrat, et al., 2021). The clear-sky shortwave direct radiative effects

(SDREs) are defined as the anomalies from the shortwave radiation field without aerosols. The SDREs have been investigated based on the numerical models, and satellite and ground-based measurements. A number of measurement-based approaches estimates the SDRE at the top of the atmosphere (TOA) to be $–5.5 \pm 0.2$ Wm$^{-2}$ over the ocean and $–4.9 \pm 0.3$ Wm$^{-2}$ over the land (Yu et al., 2006). Since the aerosol vertical profile affects the SDRE at TOA, the aerosol vertical profiles derived from the CALIOP have been considered in the evaluation of the SDREs (e.g., Oikawa et al., 2018). Furthremore, the impacts of

aerosols on the atmospheric heating rate are estimated using the aerosol vertical profiles (Korras-Carraca et al., 2019). These studies estimates the SDREs for total aerosols. In this study, the clear-sky SDREs at the top and bottom of the atmosphere and the impacts on the heating rate for each aerosol component are estimated, based on the CALIOP-MODIS retrievals.

This article is organized as follows. The CALIOP and MODIS observation data used for the retrievals are described in Sect. 2. The retrieval algorithm and the SDRE calculation method are described in Sect. 3. The uncertainties in the retrieval

results are evaluated by using simulated CALIOP and MODIS observation data in Sect. 4. The global three-dimensional distribution of aerosol compositions and the shortwave direct radiative forcing in 2010 are analysed in Sect. 5. All of the results are summarized in Sect. 6.

## 2 Data

### 2.1 Input of the CALIOP-MODIS retrievals

The CALIOP-MODIS retrieval is applied to only the clear sky (cloud-free) data of the CALIOP and MODIS observations. We made a clear-sky match-up data set of CALIOP attenuated backscatter coefficients, MODIS radiances, surface albedo, and meteorological data acquired along the orbital track of A-train satellites, which includes the CALIPSO and Aqua satellites. The CALIOP data comprise the attenuated backscatter coefficients ($\beta$) at 532 and 1064 nm and the total (or volume) depolarization ratio ($\delta$) at 532 nm in the CALIPSO Lidar Level 1B Version 4 data product (Getzewich et al., 2018; Kar et al.,

2018; Vaughan et al., 2019). The horizontal resolution of the original $\beta$ data set is 333 m, and the vertical resolution is 30 m for $\beta$ at 532 nm, and 60 m for $\beta$ at 1064 nm. Since the resolutions are different by the measurements, we created a clear sky data set with horizontal resolution of 1 km, and vertical resolutions of 120 m from -0.5 to 20.2 km altitudes and 180 m from 20.2 to 30.1 km altitudes by the following procedure. Firstly, we collected the clear sky CALIOP observations discriminated as clear air, tropospheric aerosol, and stratospheric aerosol by the vertical feature mask (VFM) product of CALIPSO Lidar

Level 2 Version 4 (Kim et al., 2018; Lie et al., 2019). The VFM product describes layer classification information and provides a cloud–aerosol discrimination (CAD) score, which is the confidence level for cloud/aerosol classification. CAD can range from –100 to +100, where positive (negative) values indicate clouds (aerosols). A higher absolute value indicates greater confidence in the classification result. In this study, we used aerosol/cloud classification results with a CAD score greater than 70 for quality assurance (Liu et al., 2009). Secondly, the clear sky CALIOP observations at our defined horizontal and vertical

coordinates were obtained by running mean with the horizontal window of 10 km and vertical windows of 120 m and 180 m. The signal noises of the CALIOP observations are reduced by the running mean.

We used Aqua MODIS Level 1B Calibrated Radiances (MYD02SSH, Collection 6.0) in bands 1 (620–670 nm) and 2 (841–876 nm) with along- and across-track resolutions of 5 km. To exclude cloud-contaminated observations, we used the Level 2 Cloud Mask Product (MYD35_L2, Collection 6.0; Ackerman et al., 2010). We used the black- and white-sky albedo

of MCD43C3 Collection 6.0 (Schaaf et al., 2002; Wang et al., 2018) for the land surface reflection in the forward calculation of MODIS observations (Sect. 3.1.2.3). The clear sky radiances and albedos at the nearest pixel within a 10-km range from the near-nadir measurements (~3° off nadir) of CALIOP were selected for retrieval.

As ancillary data for the forward calculations of CALIOP and MODIS observations, we used pressure, temperature, relative humidity, ozone concentration, and ocean surface wind speed from the MERRA-2 reanalysis data product (Gelaro et

al., 2017). The ocean surface wind speed was used in calculating the ocean surface reflection in the forward model of the MODIS observations.

### 2.2 Data for comparison of retrieval results

The results of the CALIOP-MODIS retrievals in 2010 are compared with the CALIPSO and MODIS standard products and AERONET products in Sect. 5. The CALIPSO standard product comprises the monthly means of $\tau_a$ and $\alpha_a$ in the cloud free

daytime data set of the CALIPSO Lidar Level 3 Tropospheric Aerosol product Version 4 (Tackett et al., 2018), which has longitudinal, latitudinal, and vertical resolutions of 5°, 2°, and 60 m, respectively. The MODIS standard product comprises the monthly means of $\tau_a$ in the MYD08_M3 Collection 6.1 Aqua Atmosphere Monthly Global Product (Platnick et al., 2015), with longitudinal and latitudinal resolutions of 1°. The annual means were calculated from the monthly means. The AERONET products comprise $\tau_a$, $\omega_0$, $g$, and fine and coarse mode radii in the level 2 almucantar retrievals of the version 3 inversion data product (Giles et al., 2019; Sinyuk et al., 2020).

## 3 Methods

### 3.1 Retrieval methods

#### 3.1.1 Retrieval procedure

Figure 1 is a flow diagram of the retrieval procedures. The vertical profiles of the dry volume concentrations ($V_{dry}$) of WS, LA, DS, and SS, and the columnar values of the dry median radii ($r_{m,dry}$) of the fine (WS and LA) and coarse (DS) particles are optimized to each CALIOP and MODIS data pair. $V_{dry}$ is defined as the volume of aerosols at a relative humidity of 0 % per unit atmospheric volume, and $r_{m,dry}$ is defined as the median radius of aerosols at a relative humidity of 0 %. $r_{m,dry}$ of SS is given by a parameterization that uses the ocean surface wind speed (Erickson and Duce, 1988). Only the vertical layers discriminated as aerosols in the VFM data are targeted for retrieval, and the CALIOP-MODIS retrieval is conducted for only clear sky data in the daytime. If clouds are detected in the VFM data, the CALIOP-MODIS retrieval is not conducted.

Inversion is conducted by the optimal estimation technique developed by Kudo et al. (2016). The state vector is optimized simultaneously to the measurements and a priori constraints by minimizing the following objective function:

$$f(x) = \left(\ln(y^{obs}) - \ln(y(x))\right)^T (W^2)^{-1} \left(\ln(y^{obs}) - \ln(y(x))\right) + y_a(x)^T (W_a^2)^{-1} y_a(x), \qquad (1)$$

where $x$ is the state vector to be optimized and is comprised of $V_{dry}$ for WS, LA, DS, and SS, and $r_{m,dry}$ for the fine (WS and LA) and coarse (DS) particles, vector $y^{obs}$ represents the CALIOP and MODIS measurements, vector $y(x)$ represents the calculations by the forward models corresponding to $y^{obs}$, $W^2$ is the covariance matrix of $y$, vector $y_a(x)$ gives the a priori constraints for $x$, and $W_a^2$ is an associated covariance matrix. The forward calculations of the optical properties from $V_{dry}$ and $r_{m,dry}$ for WS, LA, DS, and SS are described in Sect. 3.1.2.1. The forward models of the CALIOP and MODIS observations from the aerosol optical properties are described in Sects. 3.1.2.2 and 3.1.2.3. The details of the CALIOP-MODIS retrieval and the a priori constraints are described in Sect. 3.1.3. The minimization of $f(x)$ is conducted by an iterative algorithm, with logarithmic transformation applied to $x$ and $y$ for stable and fast convergence of the iteration. Because the CALIOP measurements can have negative values caused by large signal noise, CALIOP measurements were transformed by $\ln(y - y_{min})$, where $y_{min}$ is a possible minimum value of $y$. The best solution of $x$, which minimizes $f(x)$, is searched by the iteration of $\ln(x_{k+1}) = \ln(x_k) + \alpha d$, in $\ln(x)$ space, where vector $d$ is determined by the Gauss-Newton method, and the

scalar $\alpha$ is determined by a line search with the Armijo rule. The convergence criterion for the iteration is that the difference between $f(\boldsymbol{x}_k)$ and $f(\boldsymbol{x}_{k+1})$ is smaller than the given threshold for two consecutive times.

**3.1.2 Forward models**

**3.1.2.1 Forward model of aerosol physical and optical properties**

We assumed that the aerosols consisted of four components: WS, LA, DS, and SS. Their physical and optical properties at
relative humidities of 0 and 80 % are summarized in Table 1. WS and LA are small particles with small $g$. DS and SS are large particles with large $g$. LA and DS are light-absorbing particles and have small $\omega_0$. WS and SS have large $\omega_0$.

WS was assumed to be a mixture of sulfates, nitrates, organic and water-soluble substances (Hess et al., 1998). Their shape was assumed to be spherical, and their refractive index was defined from the OPAC database (Hess et al., 1998). We considered WS to grow hygroscopically and used the dependencies of particle size and refractive index on relative humidity
given in the OPAC database.

BC particles are emitted into the atmosphere by incomplete combustion of fossil fuels, biomass, and biofuels. The freshly emitted BC particles are generally externally mixed with the other particles and are in a hydrophobic state (Weingartner et al., 1997). These particles are gradually internally mixed by aging processes (condensation, coagulation, and/or photochemical oxidation process) in the atmosphere and become hydrophilic by coating with water-soluble compounds
(Oshima et al., 2009). We defined LA as an internal mixture of BC and WS, and introduced the core-grey shell (CGS) model (Kahnert et al., 2013). CGS model has a spherical shape with a BC core and a shell consisting of a homogeneous mixture of WS and BC. The optical properties of CGS model are better representations of a realistic encapsulated aggregate model than the internally homogeneous mixture model obtained by using the Maxwell Garnett mixing rule (MG; Maxwell Garnett, 1904) and the core-shell (CS) model. The optical properties ($\alpha_a$, $\omega_0$, $g$, and lidar ratio [$S_a$]) of CGS have values between those of
the CS and MG models (Table 1). Kahnert et al. (2013) defined a CGS model as a mixture of BC and sulfate, but we used WS instead of sulfate in our definition. The details of the application of the CGS model are described in the Appendix. The refractive index of BC was defined from the measurements of Chang and Charalampopoulos (1990). The hygroscopic growth of LA particles was considered because the WS mixed in the shell are hydrophilic. We used the dependencies of the volume and refractive index of WS on the relative humidity in the OPAC database for the shell of LA particles. In general, the volume
fraction of BC in an internally mixed particle changes spatiotemporally due to the different emission sources and the aging processes (e.g., Moteki, et al., 2007), but it is difficult to optimize the BC volume fraction in the CALIOP-MODIS retrieval. Therefore, we fixed the BC volume fraction at 30 % of the total (BC+WS) volume, which is within the range of values observed by the A-FORCE aircraft campaign in East Asia (Matsui et al., 2013). Because there are large uncertainties in the particle models and the BC volume fraction, we conducted sensitivity tests using the different particle models (CGS, CS, and MG) and
BC volume fractions (15 and 30 %) (Sect. 5).

The Voronoi particle model (Ishimoto et al., 2010) was used for DS in this study. Based on electron microscope observations, the shape of the Voronoi particle model was created by a spatial Poisson-Voronoi tessellation. As an optional model, the spheroid particle model of Dubovik et al. (2006) was also introduced in the retrieval. The particle depolarization ratio ($\delta_a$) of a spheroid particle is less than that of a Voronoi particle (Table 1). We therefore conducted a sensitivity study of the two particle models (Sect. 5). The refractive index of DS was obtained from the database of Aoki et al. (2005); this database was created from in situ measurements of dust samples in the Taklimakan Desert, China.

SS particles were assumed to be spherical, and the refractive index in the OPAC database was used. Hygroscopic growth of SS was also considered, and the particle size and refractive index were changed depending on the relative humidity. In retrievals over the ocean, four components (WS, LA, DS, and SS) were considered, but SS was ignored in retrievals over land.

Each component was assumed to have a lognormal size distribution, and hygroscopic growth was considered by including a growth factor as follows:

$$\frac{dV(r,RH)}{d\ln r} = \frac{V(RH)}{\sqrt{2\pi}\sigma} \exp\left[-\frac{1}{2}\left(\frac{\ln r - \ln r_m(RH)}{\sigma}\right)\right], \quad (2a)$$

$$r_m(RH) = GF(RH)r_{m,dry}, \quad (2b)$$

$$V(RH) = GF(RH)^3 V_{dry}, \quad (2c)$$

where $r$ is radius, $V$ is total volume, $r_m$ is median radius, $\sigma$ is the standard deviation, $RH$ is relative humidity, $GF$ is the growth factor. The standard deviation is fixed at 0.45 for WS and LA, and at 0.8 for DS and SS. These values are slightly larger than those of AERONET retrievals in worldwide locations (Dubovik et al., 2002). $r_{m,dry}$ of fine (WS and LA) and coarse (DS) particles were parameters to be optimized. Here, $r_{m,dry}$ of WS and LA were assumed to be the same. $r_{m,dry}$ of SS was determined by the following relationship between the ocean surface wind speed and the mass-mean radius for a relative humidity of 80 % (Erickson and Duce, 1988):

$$mmr = 0.422u + 2.12, \quad (3)$$

where $mmr$ is the mass-mean radius and $u$ is the ocean surface wind speed. The mass-mean radius is defined as the ratio of the fourth moment of the radius with respect to the number size distribution to the third moment (Lewis and Schwartz, 2004). $r_{m,dry}$ was calculated from $mmr$ by using the lognormal size distribution obtained by Eq. (2). The growth factor $GF$ for WS, the LA shell, and SS were obtained from the OPAC database.

To reduce the computational time, we constructed the lookup tables of $\alpha_a$, $\omega_0$, and the phase matrix for each model using the above-mentioned particle models and size distributions. The inputs of the lookup tables were $V_{dry}$ and $r_{m,dry}$ of WS, LA, DS, and SS, and relative humidity. The outputs were $\alpha_a$, $\omega_0$, the phase matrix, and the size distribution of each component at the input relative humidity. Finally, $\alpha_a$, $\omega_0$, phase matrix, $g$, $S_p$, $\delta_p$, and size distribution of total aerosols (WS+LA+DS+SS) were calculated according to the external mixture. These optical properties are used in the forward models of CALIOP and MODIS observations.

### 3.1.2.2 Forward model of CALIOP observations

We constructed a forward model to calculate $\beta$ at 532 and 1064 nm and $\delta$ at 532 nm from the vertical profiles of $\alpha_a$, $S_a$, and $\delta_a$ by the following lidar equations:

$$\beta_{co}(\lambda,z) = \left(\frac{\alpha_m(\lambda,z)}{S_m(\lambda,z)}\frac{1}{1+\delta_m(\lambda,z)} + \frac{\alpha_a(\lambda,z)}{S_a(\lambda,z)}\frac{1}{1+\delta_a(\lambda,z)}\right)\exp\left\{-2\int_{z'}^{TOA}\left(\alpha_m(\lambda,z') + \alpha_a(\lambda,z')\right)dz'\right\}, \qquad (4)$$

$$\beta_{cr}(\lambda,z) = \left(\frac{\alpha_m(\lambda,z)}{S_m(\lambda,z)}\frac{\delta_m(\lambda,z)}{1+\delta_m(\lambda,z)} + \frac{\alpha_a(\lambda,z)}{S_a(\lambda,z)}\frac{\delta_a(\lambda,z)}{1+\delta_a(\lambda,z)}\right)\exp\left\{-2\int_{z'}^{TOA}\left(\alpha_m(\lambda,z') + \alpha_a(\lambda,z')\right)dz'\right\}, \qquad (5)$$

$$\beta(\lambda,z) = \beta_{co}(\lambda,z) + \beta_{cr}(\lambda,z), \qquad (6)$$

$$\delta(\lambda,z) = \beta_{cr}(\lambda,z)/\beta_{co}(\lambda,z), \qquad (7)$$

where $\beta_{co}$ and $\beta_{cr}$ are co- and cross-polarization components of $\beta$; $\lambda$ is wavelength; $z$ is altitude; $\alpha_m$, $S_m$, and $\delta_m$ are the extinction coefficient, lidar ratio, and depolarization ratio of molecular scattering; and $TOA$ is the top of the atmosphere.

### 3.1.2.3 Forward model of MODIS observations

The band 1 and 2 radiances corresponding to the MODIS observations were calculated by the PSTAR vector radiative transfer model (Ota et al., 2010). The inputs of the forward model were the vertical profiles of $\alpha_a$, $\omega_0$, and phase matrix calculated by the forward model of the aerosol optical properties. The surface reflection over the ocean was calculated from the surface wind speed by using the physical model of Nakajima and Tanaka (1983). The surface reflection over the land was assumed to be Lambert reflectance, and the actual albedo calculated from the black- and white-sky albedo of MODIS land surface products (Sect. 2.3) was used. The actual albedo from the black- and white-sky albedo was calculated by the method of Schaaf et al. (2002). Absorption of $H_2$, $O_3$, $CO_2$, $O_2$, $O_3$, and NO gases was considered in the radiative transfer calculation. The absorption coefficient was calculated by the correlated-k distribution method (Sekiguchi and Nakajima, 2008).

For rapid calculation, the response functions of bands 1 and 2 were divided to three sub-bands. The atmospheric vertical layers were assumed to consist of five vertical layers: 0–1 km, 1–3 km, 3–6 km, 6–10 km, and 10–120 km above the surface. The influence of these assumptions was evaluated by referring to radiances simulated with the 10 sub-bands and 271 vertical layers. The properties of the aerosols, surfaces, and solar zenith angles used in the simulations were the same as those used in the simulations described in Sect. 4. The relative error of the radiances was less than 1 % for bands 1 and 2.

### 3.1.3 CALIOP-MODIS retrieval

The vertical profiles of $V_{dry}$ of WS, LA, DS, and SS, and the columnar values of $r_{m,dry}$ of fine (WS and LA) and coarse (DS) particles were optimized to each CALIOP and MODIS data pair. $r_{m,dry}$ of SS was given by the parameterization using the ocean surface wind speed. The vertical profiles of $r_{m,dry}$ were not considered in this study.

DS and SS are coarse particles, and they are more sensitive to $\beta$ at 1064 nm compared with the fine particles of WS and LA. Because only DS was assumed to be non-spherical, $V_{dry}$ of DS and SS could be estimated from $\beta$ at 1064 nm and $\delta$ at 532 nm. $V_{dry}$ of WS and LA could not be independently retrieved from only $\beta$ at 532 nm. Therefore, we introduced a priori

constraints for WS and LA, as described later. The retrieval of the median radius from the satellite measurements is highly challenging, but Kaufman et al. (2003) have shown that the effective radius can be estimated from the wavelength dependencies of $\beta$ measurements at 532 and 1064 nm, and the radiance measurements at the near infrared wavelength. We conducted a similar sensitivity study to that conducted by Kaufman et al. (2003). The scattering intensity is defined as,

$$I(\theta, \lambda) = P(\theta, \lambda)\tau_{sca}(\lambda)/(4\pi), \quad (8)$$

where $\theta$ is the scattering angle, $\lambda$ is wavelength, $P$ is the normalized phase function, and $\tau_{sca}$ is the scattering coefficient. In the calculations of the phase function and scattering coefficient, a lognormal size distribution with a standard deviation of 0.4 and the refractive index of DS were used. We calculated the scattering intensities for different wavelengths, scattering angles, median radii, and particle shapes. Figure 2 shows the ratios of the scattering intensities. The scattering intensity at the scattering angle of 180° (Fig. 2a) represents lidar measurements, and the other angles (Fig. 2b, c, and d) represent MODIS measurements. For spherical and spheroidal particles, the scattering intensity ratios increase with an increase of the median radius within the ranges of 0.05–0.2 μm and 0.5–2.0 μm. The scattering intensity ratios for Voronoi particles increase with an increase of the radius over the entire radius range. These relationships indicate that the median radii of fine and coarse particles can be estimated from the spectral information of CALIOP and MODIS measurements.

The CALIOP-MODIS retrieval procedure is diagrammed in Fig. 1, and the objective function is given by Eq. (1). The state vector $\boldsymbol{x}$ consists of the vertical profiles of $V_{dry}$ of WS, LA, DS, and SS, and $r_{m,dry}$ of fine (WS and LA) and coarse (DS) particles. $r_{m,dry}$ of WS and LA were assumed to be same. $r_{m,dry}$ of SS was given by the parameterization using ocean surface wind speed. The measurement vector $\boldsymbol{y}^{obs}$ was $\beta$ at 532 and 1064 nm, $\delta$ at 532 nm, and the band 1 and 2 MODIS radiances. The forward calculation $\boldsymbol{y}(\boldsymbol{x})$ was processed by the forward models of the CALIOP (Sect. 3.1.2.2) and MODIS (Sect. 3.1.2.3) observations. The covariance matrix $\boldsymbol{W}^2$ was assumed to be diagonal, and the diagonal element of matrix $\boldsymbol{W}$ was obtained from the measurement accuracy. The measurement accuracy of $\beta$ at 532 nm of CALIOP Version 3 was estimated by comparison with airborne high-spectral-resolution lidar (HSRL) data (Rogers et al., 2011). The mean difference was 2.9 %, and the standard deviation was 20 % in the daytime. The bias of $\beta$ at 532 nm of CALIOP Version 4 was smaller than that of CALIOP Version 3 (Getzewich et al. 2018), and our data set was smoothed by calculating the running mean (Sect. 2.1); thus, the accuracy of $\beta$ at 532 nm was assumed to be 15 %. The measurement accuracies of $\beta$ at 1064 nm and $\delta$ at 532 nm were assumed to be 20 % and 50 %, respectively. Because we could not find previous reports of the measurement accuracies of $\beta$ at 1064 nm and $\delta$ at 532 nm when we started this study, we used those values greater than the standard deviations for some scenes as the measurement accuracies. We defined the diagonal elements of $\boldsymbol{W}$ for the band 1 and 2 radiances of MODIS by the following equation,

$$W = \begin{cases} 1.0, & \text{if } \tau_a \leq 0.05 \\ \exp(\alpha\ln(\tau_a) + \beta), & \text{if } 0.05 < \tau_a < 0.5, \quad (9) \\ 0.1, & \text{if } \tau_a \geq 0.5 \end{cases}$$

where $\tau_a$ at 532 nm is obtained from the result of the CALIOP retrieval (Fujikawa et al., 2020), and the slope $\alpha$ and intercept $\beta$ values were calculated from the equation $y = \exp(\alpha\ln(x) + \beta)$ and two ordered pairs of $x$ and $y$: $(x, y) = (0.05, 1.0)$,

and $(0.5, 0.1)$. We assumed that $W$ for the radiances depended on $\tau_a$, and that its range was from 0.1 to 1.0. When $\tau_a$ is small, the upward radiance at the top of the atmosphere is significantly affected by the surface reflectance. However, we used the Lambert surface reflectance in the forward model of MODIS observations, and the surface albedo was obtained from the ancillary data. Therefore, when $\tau_a$ was small, we decreased the relative contribution of the MODIS measurements to the objective function by $W$ (Eq. (9)).

The retrieval of the vertical profiles of $V_{dry}$ is significantly affected by lidar signal noise. Smoothness of the vertical profiles of $V_{dry}$ of WS, LA, DS, and SS was assumed, and an a priori smoothness constraint was introduced by using the second derivatives for the vertical profiles of $V_{dry}$:

$$y_a(\boldsymbol{x}) = \ln V_{dry}(z_{i-1}) - 2\ln V_{dry}(z_i) + \ln V_{dry}(z_{i+1}), \qquad (10)$$

where $z_i$ is altitude. The vertical variation of $V_{dry}$ was limited by minimizing Eq. (10). The covariance matrix $\boldsymbol{W}_a^2$ in Eq. (1) was assumed to be a diagonal matrix, and the values of the diagonal elements used for the smoothness constraints were 0.2.

It is difficult to retrieve $V_{dry}$ of WS and LA independently from only $\beta$ at 532 nm. Therefore, we introduced two a priori constraints. First, the similarity of the vertical profiles of WS and LA was introduced. If the emission source of LA is the same as that of WS, for example, as with biomass-burning emissions, the vertical profile of LA would be similar to that of WS near the emission source. We assumed that the vertical profile shape of LA was similar to that of WS, and the vertical profiles of LA and WS were constrained by

$$y_a(\boldsymbol{x}) = \ln\left[V_{dry,LA}(z_i)/V_{dry,LA}(z_{i+1})\right] - \ln\left[V_{dry,WS}(z_i)/V_{dry,WS}(z_{i+1})\right], \qquad (11)$$

where $V_{dry,LA/WS}(z_i)$ is $V_{dry}$ of LA and WS at altitude $z_i$. The vertical changes in $V_{dry}$ of WS and LA approach the same values when Eq. (11) is minimized. The second constraint was the inequality of $\tau_a$ of LA and WS. In the AERONET product at worldwide locations, $\omega_0$ ranges from 0.8 to 1.0 (Dubovik et al., 2002). $\omega_0$ is about 0.96 for WS and about 0.44 for LA (Table 1), and $\omega_0$ for an external mixture of WS and LA is calculated by $\omega_0 = \left(\tau_{a,WS}\omega_{0,WS} + \tau_{a,LA}\omega_{0,LA}\right)/\left(\tau_{a,WS} + \tau_{a,LA}\right)$. Thus, $\tau_a$ of WS must be greater than that of LA. Therefore, we introduced the following log barrier function as a constraint:

$$y_a(\boldsymbol{x}) = -\ln\left(1 - \frac{\tau_{a,LA}(532nm)}{\tau_{a,WS}(532nm)}\right), \qquad (12)$$

where $\tau_{a,LA/WS}(532nm)$ are $\tau_a$ of LA and WS at 532 nm. When $\tau_{a,LA}$ approaches $\tau_{a,WS}$, Eq. (12) approaches infinity, and the objective function (Eq. (1)) also becomes infinity. The similarity and inequality constraints limited the retrieval range of LA and prevented abnormal solutions. The diagonal elements of $\boldsymbol{W}_a$ were assumed to be 1.0 for both the similarity and inequality constraints.

In addition to the above-mentioned a priori constraints, we applied an a priori constraint to $r_{m,dry}$ of fine (WS and LA) and coarse (DS) particles. The spectral dependencies of the CALIOP and MODIS measurements have information on the particle radius. However, the large noise in the CALIOP measurements affects the spectral dependencies of the CALIOP measurements, and errors in the given surface reflectance affect the forward calculation of the MODIS measurements. To avoid abnormal solutions, therefore, we constrained $r_{m,dry}$ by Eq. (13):

$$y_a(\mathbf{x}) = r_{m,dry,fine/coarse} - r_{m,dry,fine/coarse}^{a\ priori}, \qquad (13)$$

where $r_{m,dry,fine/coarse}$ is $r_{m,dry}$ of fine and coarse particles, and $r_{m,dry}^{a\ priori}$ is the a priori value. We assumed that $r_{m,dry}^{a\ priori}$ was

0.1 μm for fine particles and 2.0 μm for coarse particles. The diagonal elements $\mathbf{W}_a$ for the constraint of $r_{m,dry}$ was assumed

to be 0.2 for fine particles and 0.3 for coarse particles.

        The minimization of the objective function was based on the Gauss-Newton method (Sect. 3.1.1). This method requires the numerical derivatives of $\mathbf{y}(\mathbf{x})$, where vector $\mathbf{x}$ consists of the vertical profiles of the four aerosol components and the fine/coarse median radii, and the number of the elements is on the order of from 10 to 100. The forward calculation of the MODIS observations by PSTAR is time consuming. For more rapid calculation, therefore, we approximated the numerical

derivatives of the radiances at bands 1 and 2 for $V_{dry}$ of WS, LA, DS, and SS. First, the numerical derivative was calculated from the monochromatic radiative transfer calculation at the centre wavelengths of bands 1 and 2. Because logarithmic transformation was applied to $\mathbf{x}$ and $\mathbf{y}(\mathbf{x})$, and the best solution of $\mathbf{x}$ was searched in $\log(\mathbf{x})$ space, the numerical derivative was defined as

$$\frac{\partial \ln(y(x))}{\partial \ln(x)} = \frac{\ln(y(x+\Delta x)) - \ln(y(x))}{\ln(x+\Delta x) - \ln(x)} = \frac{\ln(y(x+\Delta x)/y(x))}{\ln((x+\Delta x)/x)}. \qquad (14)$$

$\frac{\partial \ln(y(x))}{\partial \ln(x)}$ is a relative value, and the radiances at bands 1 and 2 have no strong line absorptions. The monochromatic radiative transfer calculation for the numerical derivative is thus a good approximation. Second, the dependency of the numerical derivatives on $V_{dry}$ was investigated. Figure 3 shows an example of the approximated and reference numerical derivatives for the radiances at bands 1 and 2. The vertical profiles of WS, LA, DS, and SS used in the calculation of the numerical derivatives are shown in the first column of Fig. 3. $\tau_a$ at 532 nm used in the calculation was 0.3. The surface was the ocean, and the wind

speed was 15 m s⁻¹. The solar zenith angle was 40°. The reference numerical derivatives in the second column of Fig. 3 were calculated using the non-approximated forward model described in Sect. 3.1.2.3. The numerical derivatives mainly depend on $V_{dry}$ (the third column of Fig. 3). The altitude dependency is shown in the fourth column of Fig. 3. The altitude dependency of LA, in particular, cannot be ignored. Using these relations, we approximated the numerical derivatives by the following procedure:

(1) For each aerosol component, 10th, 30th, and 80th percentiles of $V_{dry}$ are selected. When the number of aerosol layer is few, 25th and 75th percentiles of $V_{dry}$ are selected.

(2) The numerical derivatives for the selected $V_{dry}$ are calculated for each aerosol component.

(3) The following equation is fit to the results of (2),

$$\frac{\partial \ln(y(x))}{\partial \ln(x)} = \begin{cases} (a_1 + a_2 z + a_3 z^2)V_{dry}, & \text{if three } V_{dry} \text{ are selected} \\ (a_1 + a_2 z)V_{dry}, & \text{if two } V_{dry} \text{ are selected} \end{cases}, \qquad (15)$$

where $z$ is altitude. The coefficients, $a_1$, $a_2$, and $a_3$ are determined by the fitting.

(4) The numerical derivatives at all altitudes for each aerosol component are calculated by Eq. (15).

Figure 3 shows that the approximated numerical derivatives agree well with the reference values. However, the numerical derivatives of WS and SS near the surface have a unique behaviour (see the second and fourth columns of Fig. 3), and our method could not approximate these. At present, we are unable to determine the cause of this unique behaviour.

The objective function was minimized by the method described in Sect. 3.1.1 using the approximated numerical derivatives. The outputs of the CALIOP-MODIS retrieval were the vertical profiles of $V_{dry}$ and $\alpha_a$ of WS, LA, DS, and SS, and the vertical profiles of $\alpha_a$, $\omega_0$, $g$, as well as the size distribution of total aerosols at the ambient relative humidity. Even though we introduced some approximations for more rapid calculation, the CALIOP-MODIS retrieval is still time consuming. Therefore, the CALIOP-MODIS retrieval was conducted every 5 km along the track of the CALIPSO satellte's orbit.

## 3.2 Clear sky shortwave direct radiative effect

Aerosols directly affect the radiation field within the Earth-Atmosphere system by the scattering and absorption of radiation. The aerosol-induced direct radiative effect is evaluated by the anomalies with respect to a reference state (Korras-Carraca, et al., 2021). In this study, the clear sky shortwave direct radiative effect (SDRE) was defined as the anomalies from the shortwave radiation field without aerosols, and was calculated by the following procedure. We prepared a module to calculate the aerosol optical properties ($\tau_a$, $\omega_0$, phase matrix) at any wavelengths in the solar wavelength region from the retrieved $V_{dry}$ and $r_{m,dry}$ of WS, LA, DS, and SS, and relative humidity by the forward model described in Sect. 3.1.2.1. The aerosol optical properties from 300 to 3000 nm were calculated by this module, and the clear sky SDRE of aerosols was calculated by our developed radiative transfer model (Asano and Shiobara, 1989; Nishizawa et al., 2004; Kudo et al., 2011). The solar spectrum from 300 to 3000 nm was divided into 54 intervals. Gaseous absorption by $H_2O$, $CO_2$, $O_2$, and $O_3$ was calculated by the correlated-k distribution method. We calculated the SDRE of total aerosols (WS+LA+DS+SS) and of each component (WS, LA, DS, and SS) at the top of the atmosphere (TOA) and the bottom of the atmosphere (BOA) as follows:

$$SDRE = \Delta F^{TOA/BOA} = F_{with}^{TOA/BOA} - F_{without}^{TOA/BOA}, \qquad (16)$$

where $F_{with}$ is the net flux density with the aerosol (total or each component), and $F_{without}$ is the net flux density without the aerosol (total or each component). Furthermore, we calculated the impact of aerosols on the shortwave heating rate as,

$$\Delta HR(z) = HR_{with}(z) - HR_{without}(z), \qquad (17)$$

where $HR$ is the heating rate in units of K day$^{-1}$, and $z$ is altitude.

## 4 Evaluation of retrieval uncertainties using simulation data

### 4.1 Configuration of the simulation

The uncertainties of the CALIOP-MODIS retrieval products were evaluated by applying the CALIOP-MODIS retrieval to the synthetic data of the CALIOP and MODIS observations. The synthetic data  for 16 patterns of aerosol compositions (Table 2, Fig. 4) and for different values of $\tau_a$, land and ocean surfaces, and solar zenith angles were created by the simulations using

the forward models in Sect. 3. The transport of WS, LA, and DS in the free atmosphere was considered in the biomass-burning and dust cases (Table 2). The vertical profiles for the transported aerosols were assumed to have a normal distribution (Fig. 4). The boundary layer height was 2 km, and $\alpha_a$ in the boundary layer decreased linearly with increasing altitude (Fig. 4). $r_{m,dry}$ of 0.07, 0.1, and 0.15 μm were used for WS and LA, and of 1.0, 2.0, and 4.0 μm for DS (Table 2). For $\tau_a$ at 532 nm, values of 0.05, 0.1, 0.3, 0.5, 0.7, and 1.0 were used. Three land surface types were considered, and as surface albedo at bands 1 and 2, values of 0.05 and 0.50 for grass, 0.35 and 0.41 for desert, and 0.96 and 0.88 for snow, respectively, were used. These values were taken from the ECOSTRESS Spectral Library database (https://speclib.jpl.nasa.gov/ (last access: 27 August 2022). For the ocean surface, surface wind speeds of 5, 15, and 25 m s⁻¹ were used. Solar zenith angles of 0°, 20°, 40°, and 60° were used. Random errors were added to the simulated CALIOP and MODIS observations and to the simulated surface albedo and surface wind speed data. The random errors for the CALIOP observations were less than ±15 % for $\beta$ at 532 nm, ±20 % for $\beta$ at 1064 nm, and ±50 % for $\delta$ at 532 nm. The random errors for the MODIS observations were less than ±5 % for the radiances at bands 1 and 2. The random error added to the surface albedo was less than ±0.10; this value is greater than the root mean square errors of the MOD43 albedo products: 0.07 for snow/ice surface (Stroeve et al., 2005, 2013; Williamson et al., 2016), 0.03 for agriculture, grassland, and forest (Wang et al., 2014). The random errors of surface wind speed over the ocean were considered to be less than ±5 m s⁻¹; this error is slightly larger than the root mean square errors obtained by comparing the reanalysis data with ship measurements: 2.7 to 4.10 m s⁻¹ for the National Centers for Environmental Prediction-Department of Energy reanalysis, and from 1.67 to 2.77 m s⁻¹ for the European Centre for Medium-Range Weather Forecasts Interim Re-Analysis (Li et al., 2013). Using the above conditions, the simulations of CALIOP and MODIS observations were conducted by the forward models described in Sects. 3.1.2.2 and 3.1.2.3. A total of 1152 simulations were conducted.

**4.2 Uncertainties in the retrieval products**

The retrievals of the columnar properties, $\tau_a$, $\omega_0$, and $g$ of total aerosols, $\tau_a$ of WS, LA, DS, and SS, and $r_{m,dry}$ of fine (WS and LA) and coarse (DS) particles are compared with the simulation results in Fig. 5. The plots of $\tau_a$ in Fig. 5a are aligned vertically in the lines, because we controlled the total volume of aerosols by giving $\tau_a$ at 532 nm in the simulations. Overall, the retrieval results are scattered near the one-to-one line. $\tau_a$ retrievals at 532 and 1064 nm are estimated particularly well. $\tau_a$ of WS, DS, and SS also agree with the simulated values. However, $\tau_a$ of LA is overestimated, and $\omega_0$ at 532 nm is underestimated because of the overestimation of $\tau_a$ of LA. $g$ of the CALIOP-MODIS retrieval agrees with the simulated values. $r_{m,dry}$ of fine (WS and LA) and coarse (DS) particles agree well with the simulations. Figure 6 shows box-and-whisker plots of the differences between the retrievals and simulations for different values of the simulated $\tau_a$ at 532 nm. All of the differences except for $\tau_a$ of LA and $\omega_0$ decreased with an increase of the simulated $\tau_a$, particularly in the cases with $\tau_a$ greater than 0.3. $\omega_0$ is underestimated over the entire range of simulated $\tau_a$, and $\tau_a$ of LA is overestimated. Table 3 summarizes the means and standard deviations of the differences between the retrievals and simulations, separately for the land and ocean surface results. In general, the small value of the ocean surface albedo is an ideal situation for the satellite remote sensing of

aerosols. However, the retrieval results for $\tau_a$ of WS over the ocean are worse than those over the land because SS is taken

into account, in addition to WS, LA, and DS, in the ocean surface cases. In the simulations, the random errors are added to the ocean surface wind speed. Since $r_{m,dry}$ of SS is determined by the given ocean surface wind speed and is not optimized in the CALIOP-MODIS retrieval, the random errors cause the difference of $r_{m,dry}$ of SS between the simulation and retrieval. The difference affects $\tau_a$ of SS. Since both WS and SS are less light-absorbing particles, $\tau_a$ of WS is overestimated (underestimated) when $\tau_a$ of SS is underestimated (overestimated). This opposite sign is seen in the ocean cases of Table 3.

Figure 7 shows the relative differences in $\alpha_a$ for WS, LA, DS, and SS between the retrievals and simulations. The relative differences in $\alpha_a$ for WS, LA, and DS are very large at altitudes from 3 to 5 km and from 6 to 7 km, because $\alpha_a$ is very small near the bottom and top edges of the vertical distribution of transported aerosols (see Fig. 4). The relative difference in $\alpha_a$ for WS ranges from –30 to 10 %, and it tends to be underestimated at all altitudes except for the bottom and top edges of the transported aerosol layer. The median value of the relative differences is close to 0 %. The relative difference in $\alpha_a$ for

LA tends to be overestimated and ranges from –100 to 200 %; The median value in the boundary layer is close to 0 %, but the variances are large. $\alpha_a$ of DS tends to be underestimated; the relative difference ranges from –50 to 0 %. The relative difference in $\alpha_a$ for SS tends to be overestimated; the relative error is from –40 to 40 %. Table 4 shows the means and standard deviations of these relative differences and the differences for $\alpha_a$, $\omega_0$, and $g$ of total aerosols. $\alpha_a$ of LA over the land was overestimated, and this was compensated by the underestimating $\alpha_a$ of WS and DS. Hereby, the relative difference of $\alpha_a$ for total aerosols

was small, about –4 %. In the ocean cases, $\alpha_a$ of LA and SS was overestimated, and this was compensated by underestimating $\alpha_a$ of WS and DS. Similar to the results for the columnar properties, the results of $\alpha_a$ the ocean are worse than those over the land.

Overall, the uncertainties in the retrieval results over the land are smaller than those over the ocean. The retrieval results become better in the larger $\tau_a$ cases. The CALIOP-MODIS retrievals tend to overestimate the amount of LA, and $\omega_0$

is underestimated. The retrieval of $r_{m,dry}$ is a challenging problem, but $r_{m,dry}$ of fine (WS and LA) and coarse (DS) particles are estimated well.

## 5 Retrieval results from the CALIOP and MODIS observations in 2010

### 5.1 Global 3D distribution

The annual means of $\tau_a$ and $\alpha_a$ in the CALIOP-MODIS retrievals for 2010 are compared with the CALIPSO and MODIS

standard products in Fig. 8. The grid resolutions are 5° latitude by 2° longitude for the CALIOP-MODIS retrieval and the CALIPSO standard product and 1° latitude by 1° longitude for the MODIS standard product. Note that the MODIS standard product is at 550 nm, but the difference of $\tau_a$ between 532 and 550 nm is small. The horizontal distributions of $\tau_a$ are similar in all results. Large $\tau_a$ values are distributed in the middle of the Atlantic Ocean, Africa, and Western, Southern, and Eastern Asia. The global mean ± standard deviation of $\tau_a$ was 0.113 ± 0.161 for the CALIPSO standard product, 0.147 ± 0.148 for the

CALIOP-MODIS retrieval, and $0.164 \pm 0.145$ for the MODIS standard product. Thus, the global mean of the CALIOP-MODIS retrieval was between those of the CALIPSO and MODIS standard products and was close to that of the MODIS standard product. Compared with $\tau_a$ of the AERONET, the CALIPSO version 4 product has a negative bias of $-0.05 \pm 0.085$ (Kim et al., 2018), and $\tau_a$ of the merged data set of the dark target (DT) and deep blue (DB) algorithms in the Aqua MODIS collection 6.1 product has a small positive bias of 0.004 (Shi et al., 2019). Considering that the CALIOP-MODIS retrieval method used both CALIOP and MODIS observations, the retrieval result is reasonable and is better than the CALIPSO standard product.

The zonal means of $\alpha_a$ in all results showed similar distributions to the CALIPSO standard product. $\alpha_a$ was large at latitudes from 60°S to 40°S and from 0° to 30°N. The top altitude of the vertical distribution was about 5 km at latitudes from 0° to 30°N. In the CALIOP-MODIS retrieval, slightly large $\alpha_a$ were observed at altitudes from 0 to 9 km and latitudes from 70°S to 80°S, and a peak of $\alpha_a$ at altitudes from 0 to 1 km and latitudes around 70°N. These unnaturally large values in the polar regions may be attributable to cloud contamination. Additionally, since the CALIOP-MODIS retrieval is attempted to the observation data over the ice surface, it is possible that the high albedo of the ice surface result in the unnatural $\alpha_a$.

We further compared the regional distributions of $\tau_a$ with the CALIOP and MODIS standard products. In North America, South America, and Europe, the CALIOP-MODIS retrieval is close to the MODIS standard product. In the Africa, the CALIOP-MODIS retrieval is between the MODIS and CALIOP standard products, but the CALIOP standard product is largest in Western Africa, and the CALIOP-MODIS retrieval was smallest in the three products. Additionally, the famous dust source, the Bodélé depression located northeast of Lake Chad in Middle Africa (Koren et al., 2006) can be clear in the MODIS standard product but cannot be detected in the other two products. The local dust source of the Bodélé depression did not appear in the CALIOP-MODIS retrieval even though the MODIS measurements are utilized. This detection failure of the local dust source may be attributed to the sparse observations of the CALIOP in the longitude direction. In Western, Southern, and Eastern Asia, the CALIOP standard product is larger than the MODIS product, and the CALIOP-MODIS retrieval is between the two standard products. In Australia, the CALIOP-MODIS retrieval was largest. The values of $\tau_a$ in the three products are different by the regions. Kim et al. (2018) also shows the different positive and negative biases by the regions in the comparisons of the CALIOP and MODIS products. The comparisons of $\tau_a$ of the Aqua MODIS collection 6.1 products with the AERONET products also shows that the bias sign is different for the regions and the DT and DB algorithms (Sayer et al., 2019; Shi et al., 2019; Wei et al., 2020; Huang et al., 2020; Eibedingil et al., 2021; Sharma et al., 2021). The further comparisons of the CALIOP-MODSI retrieval with the L2 products of CALIOP and MODIS in the regional scale are necessary in the future.

Figure 9 shows the horizontal distributions of $\omega_0$ and $g$ of the CALIOP-MODIS retrieval. The global means of $\omega_0$ and $g$ were about $0.940 \pm 0.038$ and $0.718 \pm 0.037$. Previous studies have shown that the global mean $\omega_0$ is from 0.89 to 0.953 (Korras-Carraca et al., 2019; Kinne, 2019), and the global mean $g$ is 0.702 (Kinne, 2019). Our results are thus consistent with these previous studies. $\omega_0$ over the land was from 0.8 to 0.95 and was smaller than that over the ocean. $g$ over the land was from 0.6 to 0.75 and also smaller than that over the ocean. These differences between land and ocean are due to the presence

of SS over the ocean, because $\omega_0$ and $g$ of SS are larger than those of the other aerosol components (Table 1). In the major biomass-burning regions of the central and southern parts of South America, and the southern part of Africa, $\omega_0$ and $g$ of the CALIOP-MODIS retrieval are particularly small, from 0.85 to 0.90, and 0.65 to 0.70, respectively. These are consistent with the results of Kinne (2019). However, our retrieved $\omega_0$ is less than 0.90 over the most parts of the land area and appears to be about 0.05 smaller than $\omega_0$ of Kinne (2019). In Sect. 4, it was shown that the CALIOP-MODIS retrieval tended to underestimate $\omega_0$. The tendency to underestimate $\omega_0$ might appear in the retrieval over the land.

Figure 10 depicts the horizontal distributions of $\tau_a$ of WS, LA, DS, and SS. Note that the ranges of $\tau_a$ depicted by colour bars in Fig. 10 are different. $\tau_a$ of WS was large over South America, Africa, Western, Southern, and Eastern Asia, and the ocean. The large $\tau_a$ of WS over the ocean might include contributions from fine SS particles, and biogenic sulfate or organic compounds because a large $\tau_a$ of WS was also seen over regions where the surface wind speed is large, such as the sea around Antarctica. A large $\tau_a$ of LA was seen in South America, Middle Africa, and Southern and Eastern Asia, which are major sources of aerosols from anthropogenic and biomass-burning sources. $\tau_a$ of DS was large around the desert regions of the northern part of Africa, and Western, Southern, and Eastern Asia. Compared with those WS, LA, and DS, $\tau_a$ of SS was smaller and was uniformly distributed over the ocean, but a peak was found in Arabian Sea, where there are the strong persistent southerly and southwesterly winds from June to September (Chaichitehrani and Allahdadi, 2018), and the strong northerly winds, shamal and makran winds, from October to January (Aboobacker et al., 2021). The global mean of $\tau_a$ was $0.072 \pm 0.085$ for WS, $0.027 \pm 0.035$ for LA, $0.025 \pm 0.054$ for DS, and $0.023 \pm 0.020$ for SS, respectively. We compared the global distributions of each component with the previous studies of Kinne (2019), Gkikas et al. (2021), and Korras-Carraca et al. (2021). The global distributions of $\tau_a$ of WS, LA, and SS matched well with those of sulfate + organic, BC, and SS in Fig. 6 of Kinne (2019), and Fig. 1 of Korras-Carraca et al. (2021), respectively. Here, we compared WS of this study with sulfate + organic of Kinne (2019) and Korras-Carraca et al. (2021) because our definition of WS (Sect. 3.1.2.1) is similar to sulfate + organic. The global distribution of $\tau_a$ of DS was also consistent with those of Kinne (2019), Gkikas et al. (2021), and Korras-Carraca et al. (2021). The global mean of $\tau_a$ for the fine particle (WS + LA) in the CALIOP-MODIS retrieval is 0.097, which is greater than 0.063 for the fine particle (sulfate + organic + BC) in Kinne (2019), and 0.08 for the fine particle (sulfate + organic + BC) in Korras-Carraca et al. (2021). We compared $\tau_a$ of the fine particle because LA in this study is defined as an internal mixture of WS and BC, and is different from the pure BC defined in Kinne (2019) and Korras-Carraca et al. (2021). The global mean of $\tau_a$ for SS was 0.028 in the Kinne (2019), and 0.04 in Korras-Carraca et al. (2021). The global mean of $\tau_a$ for DS was 0.031 in Kinne (2019), 0.033 in Gkikas et al. (2021), and 0.03 in Korras-Carraca et al. (2021). Consequently, $\tau_a$ of SS and DS in the CALIOP-MODIS retrieval were slightly smaller than the previous studies, and $\tau_a$ of the fine particle is larger than the previous studies. This study is the result in 2010, but the data of Kinne (2019) is the result in 2005, and the data of Korras-Carraca et al. (2021) is the means in 1980-2019. The temporal change is one of the plausible causes for the above differences of the fine particles because the emissions of the anthropogenic aerosols have large variability (Quass et al., 2022).

Figure 11 shows the zonal means of $\alpha_a$ of WS, LA, DS, and SS. Note that the range of $\alpha_a$ depicted by colour bar in Fig. 11b is smaller than those in Figs. 11a, c, and d. The distribution of WS is almost the same as that of total aerosols (Fig. 8b and d). $\alpha_a$ of WS was largest among the four aerosol components, and $\alpha_a$ of LA was smallest. The distribution of DS is concentrated between latitudes of 0° and 50°N, and the top altitude is about 5 km. SS is distributed across all latitudes, and its top altitude is about 1 km.

Figure 12 shows $r_{m,dry}$ of WS, LA, DS, and SS particles. $r_{m,dry}$ of WS, LA, and DS are large over the land and small over the ocean. This result indicates that particle size decreases away from the source regions due to the dry deposition. $r_{m,dry}$ of SS is the result of the parameterization using the ocean surface wind speed. Because $r_{m,dry}$ of SS increases with an increase of wind speed, it is large in the midlatitudes, where cyclones caused by baroclinic instability occur frequently.

**5.2 Comparisons with AERONET products**

The CALIOP-MODIS retrieval results in 2010 were compared with the AERONET products. The CALIOP measurements are near-nadir (~3° off nadir) and include no swath observations. Most AERONET sites are far from the CALIPSO ground track. Because mesoscale variability is a common feature of lower-tropospheric aerosols (Anderson et al., 2003), Omar et al. (2013) introduced as criteria for the coincidence a CALIPSO overpass with an AERONET site ±2 h and within a 40-km radius of the AERONET site. Schuster et al. (2012) used the coincidence criteria of ±30 min, within an 80-km radius, and a CALIOP digital

elevation model surface elevation within 100 m of the AERONET site elevation. In this study, we used coincidence criteria of ±2 h, within a 40-km radius of an AERONET site, and within ±100 m of the AERONET site elevation. We thus compared the means of CALIOP-MODIS retrievals satisfying these spatial criteria with the means of AERONET retrievals within ±2 h. A total of 91 samples for 51 AERONET stations (Fig. S1) met these criteria. The columnar properties of $\tau_a$ at 532 and 1064 nm, $\omega_0$ at 532 nm, $g$ at 532 nm, and the fine and coarse median radii of the volume size distribution at the ambient relative humidity

were compared (Fig. 13). The AERONET optical properties at 532 and 1064 nm were calculated from the data at the AERONET wavelengths of 440, 500, 675, and 870 nm by linear interpolation and extrapolation in a log–log space. We used $\tau_a$ directly derived from the sun-direct measurements, and $\omega_0$, $g$, and the fine and coarse median radii of the volume size distribution are the results of the almucantar retrievals in the AERONET level 2 product. The fine and coarse median radii of the CALIOP-MODIS retrieval data were calculated from the column-integrated volume size distribution by the same method

as that used for AERONET data (Dubovik et al., 2002).

        $\tau_a$ at 532 and 1064 nm of CALIOP-MODIS retrievals agreed well with those of AERONET; the slopes of the relationships were almost 1.0. The means and standard deviations of the relative differences between the CALIOP-MODIS retrievals and AERONET products were 9 ± 80 % for $\tau_a$ at 532 nm, and –1 ± 48 % for $\tau_a$ at 1064 nm.

        $\omega_0$ plots were fewer than those of the other parameters. $\omega_0$ retrieved from the sun-sky photometry has high

uncertainty when AOD is small (Sinyuk et al. 2020; Kudo et al., 2021), and the AERONET Level 2 product does not provide the retrieved SSA when $\tau_a$ at 440 nm is less than 0.4. The coefficient of determination in $\omega_0$ comparison was small, and the

CALIOP-MODIS retrievals were underestimated. The mean ± standard deviation of the absolute differences of $\omega_0$ at 532 nm was −0.04 ± 0.04. The coefficient of determination for $g$ comparison was also small, and the CALIOP-MODIS retrievals were slightly underestimated. The mean ± standard deviation of the absolute differences of $g$ at 532 nm was −0.04 ± 0.05. The coefficient of determination for the fine median radius of the CALIOP-MODIS retrieval was small, 0.015. However, the fine median radii of both the CALIOP-MODIS retrieval and the AERONET product lay in the same range from 0.1 to 0.2 μm, and the mean ± standard deviation of the absolute differences was 0.01 ± 0.03 μm. The comparison of the coarse median radius also showed a small coefficient of determination, 0.054. However, the mean ± standard deviation of the absolute difference was small, 0.35 ± 0.62 μm, because the coarse median radii of the CALIOP-MODIS retrieval and the AERONET product lay in a similar range from 1.0 to 3.5 μm.

In summary, $\tau_a$ at 532 and 1064 of the CALIOP-MODIS retrievals showed good agreement with those of the AERONET products. $\omega_0$, $g$, and fine and coarse median radii were not retrieved well, but their values were not far from those of the AERONET products. The vertical profile of $\alpha_a$ was not compared with ground-based measurements in this study. In the future, we will compare the vertical profile of $\alpha_a$ with HSRL and Raman lidar measurements in the AD-Net (Nishizawa et al., 2017; Jin et al., 2022).

## 5.3 Influences of particle models

The assumed particle model is important in the retrieval of aerosols. We therefore investigated how different particle models influenced the retrievals by comparing the results when the spheroid particle model for DS was used in the retrievals instead of the Voronoi particle model. Figures S2 and S3 show the differences of the retrieval results between the spheroid and Voronoi particle models. $\tau_a$ of DS for the retrieval with the spheroid model was greater than that for the retrieval with the Voronoi model (Fig. S2). Because $\delta_a$ of the spheroid particle model is smaller than that of the Voronoi model (Table 1), a large amount of DS was required to fit $\delta$ calculated by the forward model to $\delta$ measurements when the spheroid model was used. $\tau_a$ of WS and LA was decreased to compensate for the increase in $\tau_a$ of DS. The retrieved $r_{m,dry}$ of DS was decreased (Fig. S3) by as much as about 0.6 μm in the heavy dust regions of Africa and western Asia. In Sect. 3.1.3, we showed that the median radius can be estimated from the spectral information of the scattering intensity. The scattering intensity ratio for spheroid particles changes from 0.8 to 3.0 in the range of the median radius from 1.0 to 5.0 μm, whereas the ratio of the scattering intensity for Voronoi particles changes from 0.8 to 2.6 in the median radius range from 1.0 to 5.0 μm (Fig. 2a). Since the scattering intensity ratio for spheroid particles is larger than that for Voronoi particles in the range from 1.0 to 5.0 $\mu$ m, the retrieved $r_{m,dry}$ of DS in the retrieval with the spheroid particle model was smaller than that in the retrieval with the Voronoi model. $r_{m,dry}$ of WS and LA were not influenced by the particle model used for DS.

The fixed volume fraction of BC is one of the assumptions associated with large uncertainties in this study. We therefore conducted the retrieval using LA with a BC volume fraction of 15 % instead of 30 %. Table 5 and Fig. S4 shows the difference in the retrieval results between BC volume fractions of 15 % and 30 %. $\tau_a$ of WS and LA were slightly decreased

(Fig. S4b and c). The decrease in the global mean $\tau_a$ was less than 0.01, but the decrease was large, up to 0.03, in Africa and
western, southern, and eastern Asia. These results can be explained by the changes in $\omega_0$ and $S_a$. $\omega_0$ of LA with a BC fraction
of 15 % is greater than that with a BC fraction of 30 %, and $S_a$ of LA with a BC fraction of 15 % is smaller than that with a
BC fraction of 30 % (Table 1). Larger $\omega_0$ and smaller $S_a$ induces an increase in the values of the MODIS radiances and the
CALIOP backscatter coefficients calculated by the forward models. As a result, smaller $\tau_a$ and $\alpha_a$ are retrieved. The influence
of the BC volume fraction on the retrieved $\tau_a$ of DS and SS (Fig. S4d and e) and on $r_{m,dry}$ of the fine (WS and LA) and coarse
(DS) particles was negligible (Table 5).

We also investigated the differences in retrievals when the CGS, CS, and MG models were used. The impacts on the
retrieved $\tau_a$ are summarized in Table 5. The retrieval using MG slightly increased $\tau_a$ of LA because of a slightly large $S_a$
(Table 1). Conversely, the retrieval using CS decreased $\tau_a$ of LA because $S_a$ of CS was smaller than that of CGS (Table 1).
Different mixture models affected only the WS and LA retrievals, and the impact on the global mean $\tau_a$ was less than 0.01.

**5.4 Clear sky shortwave direct radiative forcing**

The clear sky SDRE values of aerosols at the bottom and top of the atmosphere and the impacts of aerosols on the atmospheric
heating rate were calculated from the retrieval results described in Sect. 5.1. The annual mean of the SDRE at the top of the
atmosphere was −4.99 ± 3.42 W m$^{-2}$ (Fig. 14). Korras-Carraca et al. (2019) summarized the SDRE obtained by previous studies
based on CALIOP and MODIS observations and chemical transport models, and Korras-Carraca et al. (2021) shows the SDRE
of the 40-year climatology of the MERRA-2 reanalysis data. These previously obtained SDRE values ranged from −2.6 to
−7.3 W m$^{-2}$ for $\tau_a$ from 0.074 to 0.18, and for $\omega_0$ from 0.89 to 0.97. Our results thus are in the range of previously obtained
values. The horizontal distribution of the SDRE was also similar to those of previous studies (Korras-Carraca et al., 2019;
2021), and positive forcing was observed over desert and snow/ice surfaces with a large surface albedo. An advantage of this
study is that the SDRE of each aerosol component was determined. The global mean SDRE of WS was −2.99 ± 1.49 W m$^{-2}$,
whereas the global mean SDRE of LA was 0.22 ± 0.94 W m$^{-2}$, and the SDRE of LA was positive in almost all regions. The
global mean SDRE of DS was −0.93 ± 1.32 W m$^{-2}$, but the SDRE of DS was positive over desert and snow/ice surfaces. The
SDRE of SS was negative worldwide at −0.96 ± 0.62 W m$^{-2}$. Korras-Carraca et al. (2021) also shows the global mean of the
SDRE at the top of the atmosphere for each component, and the SDRE is −1.88 W m$^{-2}$ for sulfate, −0.73 W m$^{-2}$ for organic
carbon, 0.19 W m$^{-2}$ for BC, −0.83 W m$^{-2}$ for DS, and −1.62 W m$^{-2}$ for SS. The SDRE of WS in this study is close to the SDRE
of sulfate + organic carbon of Korras-Carraca et al. (2021). However, note that a simple addition of the SDRE for sulfate and
organic carbon does not accurately represent the SDRE of sulfate + organic carbon because the SDRE responds nonlinearly to
the changes in the aerosol optical properties. The SDRE of LA and DS are also consistent with those of Korras-Carraca et al.
(2021). Only the SDRE of SS in this study was smaller than that of Korras-Carraca et al. (2021) because $\tau_a$ of SS of Korras-
Carraca et al. (2021) is 0.04 and is greater than this study.

The SDRE at the bottom of the atmosphere was negative in all regions, and the global mean was −13.10 ± 9.93 W m$^{-2}$ (Fig. 15). Previously reported values ranged from −10.7 to −6.64 W m$^{-2}$ (Korras-Carraca et al., 2019; 2021). The CALIOP-MODIS retrieval result was more negative than the previous study results. The global mean of the SDRE for each component was −4.18 ± 2.98 W m$^{-2}$ for WS, −4.66 ± 3.88 W m$^{-2}$ for LA, −2.86 ± 4.28 W m$^{-2}$ for DS, and −1.12 ± 0.73 W m$^{-2}$ for SS. Although $\tau_a$ of LA was smaller than $\tau_a$ of WS (Fig. 10), the SDRE of LA was largest. Furthermore, whereas $\tau_a$ of DS was

comparable to that of SS, the SDRE of DS was larger than that of SS. The small $\omega_0$ of LA and DS decreases the diffuse irradiance reaching the surface, with the result that the SDRE at the bottom of the atmosphere becomes large (Kudo et al., 2010b). Korras-Carraca et al. (2021) shows the global mean of the SDRE at the bottom of the atmosphere is −1.86 W m$^{-2}$ for sulfate, −0.91 W m$^{-2}$ for organic carbon, −0.72 W m$^{-2}$ for BC, −1.98 W m$^{-2}$ for DS, and −1.74 W m$^{-2}$ for SS. The global distributions of the SDRE for each component in this study were consistent with those of Korras-Carraca et al. (2021), but the

magnitudes of the SDRE were significantly different, particularly in the results of the fine particles (WS, and LA). The value of $\tau_a$ for the fine mode is 0.099 in the CALIOP-MODIS retrieval (WS+LA), and 0.08 in Korras-Carraca et al. (2021) (sulfate+organic+BC). The difference of $\tau_a$ is small. Since small value of $\omega_0$ results in large SDRE at the bottom of the atmosphere, the underestimation of $\omega_0$ in the CALIOP-MODIS retrieval (Sects. 4, and 5.2) is a possible cause. Further studies regarding to the differences of the aerosol optical properties and the configuration of the radiative transfer model are necessary

in the future. Figure 16 shows the zonal means of the aerosol impacts on the heating rate. The vertical distribution of the impacts of the total aerosols corresponds to the distribution of $\alpha_a$ (Fig. 8). The maximum heating rate was about 0.5 K day$^{-1}$. Korras-Carraca et al. (2019) also found that the aerosol impact on the heating rate was large in the boundary layer, with a maximum value of about 0.5 K day$^{-1}$. LA had the largest impact on the heating rate because of its small $\omega_0$, despite its small $\alpha_a$ (Fig. 11). The values at altitudes from 0 to 9 km and latitudes from 70°S to 80°S in Figs. 16a, b, c, and d were unnatural.

These unnatural values correspond to the unnatural $\alpha_a$ described in Sect. 5.1. Cloud contamination and high surface albedo of ice are possible causes. We showed that the CALIOP-MODIS retrieval overestimates the amount of LA and underestimates $\omega_0$, and the SDRE at the bottom of the atmosphere is more negative than the previous studies. Considering these factors, the impacts of LA on the heating rate might be overestimated. The aerosol-induced changes in the atmospheric heating rate affect the atmospheric stability and regional dynamics (Yu et al., 2002; Huang et al., 2014; Kudo et al., 2018). Improvement of the

retrieving LA and SSA is necessary.

To summarize, the SDRE values calculated from the CALIOP-MODIS retrievals are consistent with those of previous studies. However, SDRE values at the bottom of the atmosphere were larger than in the previous studies. LA had a significant impact on the SDRE at the top and bottom of the atmosphere and on the heating rate. However, the CALIOP-MODIS retrievals tended to overestimate the amount of LA and underestimate $\omega_0$. Thus, the retrieval of LA needs to be improved in the future.

## 6 Summary and conclusions

We developed the CALIOP-MODIS retrieval method for the observation of the global three-dimensional distribution of aerosol composition. The CALIOP-MODIS retrieval optimizes the aerosol composition to both CALIOP and MODIS observations in the daytime. In this study, aerosols were assumed to consist of four components, WS, LA, DS, and SS. The CALIOP-MODIS retrieval optimizes the vertical profiles of $V_{dry}$ of the four components and $r_{m,dry}$ of fine (WS and LA) and coarse (DS) particles to the CALIOP and MODIS observations. The outputs of the CALIOP-MODIS retrieval are the vertical profiles of $\alpha_a$, $\omega_0$, and $g$ of total aerosols (WS+LA+DS+SS) as well as $\alpha_a$ of WS, LA, DS, and SS, and their columnar integrated or mean values.

The uncertainties in the retrieval products were evaluated by using simulated data of the CALIOP and MODIS observations. Simulations were conducted for 16 aerosol vertical profile patterns by assuming the actual scenes in the daytime, including transport of dust, biomass-burning, and polluted dust with different $\tau_a$ for total aerosols, different land (grass, desert, and snow) and ocean (different values of surface wind speed) surfaces, and different solar zenith angles. Random errors were also added to the CALIOP and MODIS observations, surface albedo, and surface wind speed. Overall, the performance of the CALIOP-MODIS retrievals was good. The retrieval results in the case of land surfaces were better than those for the ocean surface, because three components, excluding SS, were retrieved over the land surface, whereas four components were retrieved over the ocean surface. The retrieval results became better when $\tau_a$ was increased. However, the amount of LA tended to be overestimated regardless of $\tau_a$ and land or ocean surface; hence, $\omega_0$ tended to be underestimated.

Daytime observation data of CALIOP and MODIS in 2010 were analysed by the CALIOP-MODIS retrieval. The global mean of $\tau_a$ for total aerosols was $0.147 \pm 0.148$. Comparison with the CALIPSO and MODIS standard products showed that $\tau_a$ of the CALIOP-MODIS retrieval was between those of the CALIPSO and MODIS standard products and was close to the MODIS standard product. Since the previous studies show the CALIPSO standard product tends to underestimate $\tau_a$ and the MODIS standard product has a small positive bias, the results obtained are reasonable and are better than the CALIOP standard product. The horizontal distribution of $\tau_a$ for total aerosols in the CALIOP-MODIS retrieval was generally similar to the distributions in the CALIPSO and MODIS standard products. However, there were some regional differences between the CALIOP-MODIS retrieval, and the CALIOP and MODIS standard products. The vertical distribution of the CALIOP-MOIDS retrieval was also similar to that in the CALIPSO standard product. Additionally, the an unnaturally large $\alpha_a$ due to cloud contamination and high surface albedo of ice was found in both polar regions in the CALIOP-MODIS retrievals.

The global means of $\tau_a$ for each component were $0.072 \pm 0.085$ for WS, $0.027 \pm 0.035$ for LA, $0.025 \pm 0.054$ for DS, and $0.023 \pm 0.020$ for SS. The global distributions of WS, LA, DS, and SS in this study were consistent with those for sulfate + organic, BC, DS, and SS in the previous studies of Kinne (2019), Gkikas et al. (2021), and Korras-Carraca et al. (2021). The global means of $\tau_a$ for DS and SS in this study were slightly smaller than those of Kinne (2019), Gkikas et al. (2021), and Korras-Carraca et al. (2021), whereas the global mean of $\tau_a$ for the fine particle (WS + LA) was greater than the fine particles (sulfate + organic + BC) of Kinne (2019), and Korras-Carraca et al. (2021). The data of Kinne (2019) is the result in 2005, and

the data of Korras-Carraca et al. (2021) is the means in 1980 to 2019, but this study is the result in 2010. Since anthropogenic aerosol emissions have large temporal variability (Quass et al., 2022), we need to compare the result in the same period for the further investigation.

Using the retrieval results for 2010, $\tau_a$, $\omega_0$, $g$, and fine and coarse median radii of the CALIOP-MODIS retrievals were compared with the corresponding AERONET products. $\tau_a$ at 532 and 1064 nm of the CALIOP-MODIS retrieval agreed well with the AERONET product. The values of $\omega_0$, $g$, and fine and coarse median radii of the CALIOP-MODIS retrievals were not far from those of the AERONET products, but the variations were large and the coefficients of determination for linear regressions between the CALIOP-MODIS retrievals and the AERONET products were small. $\omega_0$ and $g$ were underestimated by about 0.04, compared with those of the AERONET.

The assumed particle model used in the retrieval causes large uncertainties. We investigated the influences of the DS shapes (Voronoi or spheroid models), the volume fractions of BC in LA (15 or 30 %), and the internal mixtures of LA (CGS, MG, and CS models). The DS shapes had large impacts on DS, WS, and LA, and the difference of global mean of $\tau_a$ was about 0.015 for DS, and less than 0.01 for WS and LA. The influences of the volume fractions and internal mixtures of LA were small, less than 0.01 for the global mean $\tau_a$ of WS and LA, but the influences cannot be ignored in the regional scale. The difference of $\tau_a$ of LA was greater than 0.01 around the central part of Africa, where is the famous biomass-burning region.

The clear sky SDRE of aerosols at the top and bottom of the atmosphere and the impact of aerosols on the heating rate was investigated using the retrievals for 2010. The SDRE values at the TOA and BOA were –4.99 ± 3.42 and –13.10 ± 9.93 W m$^{-2}$, respectively. The SDRE at the TOA is in the range of previously reported values (from –2.6 to –7.3 W m$^{-2}$). However, the SDRE at the BOA was greater than previously reported values (from –10.7 to –6.64 W m$^{-2}$). The aerosol impact on the heating rate ranged from 0.0 to 0.5 K day$^{-1}$, consistent with previously reported values. The horizontal distributions of the SDRE at the TOA and BOA, and the vertical distributions of the aerosol impacts on the heating rate were consistent with those of previous studies. An advantage of this study was that the SDRE was estimated for each aerosol component. The AOD of WS was largest among the four aerosol components: the SDRE of WS at the TOA and BOA was large, but the impact of WS on the heating rate was small because WS is a less light-absorbing particle. In contrast, $\tau_a$ of LA was small, but its SDRE at the TOA was positive in most of the world, and its SDRE at the BOA and its impact on the heating rate were largest among the four aerosol components. Thus, although the amount of LA was small, but the impact on the SDRE was significantly important. The SDRE at the TOA and BOA for each component were compared with those of Korras-Carraca et al. (2021). The distributions of the SDRE at the TOA and BOA for each component were consistent with those of Korras-Carraca et al. (2021). Furthermore, the global mean values of the SDRE at the TOA agreed well with each other. However, the SDRE at the BOA for each component in this study were greater than those of Korras-Carraca et al. (2021), particularly for the fine particles (WS and LA). A possible reason except for the configuration of the radiative transfer model would be the differences of the aerosol optical properties. $\tau_a$ for the fine particle (WS + LA) was slightly greater than the fine particles (sulfate + organic + BC) of Korras-Carraca et al. (2021), and the CALIOP-MODIS retrieval overestimates the amount of LA and underestimates SSA. The smaller SSA results in the more negative SDRE at the TOB.

Consequently, $\tau_a$ and $\alpha_a$ of the CALIOP-MODIS retrieval in 2010 showed reasonable results when compared with the CALIPSO and MODIS standard products, the AERONET products, and the previous studies. Furthermore, the SDRE values calculated from the CALIOP-MODIS retrievals were consistent with those of previous studies. However, there were some issues with the CALIOP-MODIS retrievals. $\tau_a$ of LA tended to be overestimated, and $\omega_0$ was underestimated. Because the LA has a large impact on the SDRE, the overestimation of LA should be improved in a future study. The unnaturally large $\alpha_a$ in both polar regions is also an important issue. The further investigations for the cloud discrimination and the surface albedo of the ice-covered regions are necessary. There were some differences of $\tau_a$ between the CALIOP-MODIS retrieval, and the CALIOP and MODIS standard product in the regional scale. Further validation study using the AERONET product is required. Additionally, the vertical profile of $\alpha_a$ was not validated in this study. We will compare the vertical profile of $\alpha_a$ with the ground-based measurements by HSRL and Raman lidar of the AD-Net in the future.

The Earth Clouds, Aerosol and Radiation Explorer (EarthCARE) satellite is a joint mission of the European Space Agency and the Japanese Aerospace Exploration Agency (Illingworth et al., 2015). The atmospheric lidar (ATLID) onboard EarthCARE is a linearly polarized HSRL transmitting a spectrally narrow laser beam at 355 nm. The multispectral imager (MSI) onboard EarthCARE is an imager with seven bands from visible to infrared wavelengths. We are developing the application of the CALIOP-MODIS retrieval to the ATLID and MSI observations. The lidar ratio is an optical parameter related to $\omega_0$ and can be directly retrieved from the ATLID measurements. Hereby, the EarthCARE observations may improve the retrieval of $\omega_0$ and LA. Furthermore, we plan to investigate long-term changes in the aerosol composition by using the CALIOP and MODIS observations together with ATLID and MSI observations. The results will contribute to our understanding of climate changes due to aerosols.

**Appendix A: Optimization of the core-grey shell model**

BC has a complex morphology and forms mixtures with weakly light-absorbing particles. Previous studies have developed various simplified models, such as externally mixed homogeneous spheres, an internally mixed homogeneous sphere, and the CS model. Comparison with realistic encapsulated aggregate models has shown that the externally mixed homogeneous spheres and the CS model underestimate the absorption cross section, and that the internally mixed homogeneous sphere overestimates the absorption cross section (Kahnert et al., 2012). The CGS model, developed by Kahnert et al. (2013), has a CS geometry, but compared with the original CS model with the same volume of BC and weakly light-absorbing particles, the volume fraction of the BC core to the total BC volume in a particle ($f_{\text{core}}$) is smaller than one in the CGS model, and the remaining BC ($1 - f_{\text{core}}$) is homogeneously mixed with weakly light-absorbing particles in the shell. The Maxwell Garnett mixing rule is used for the homogeneous mixing in the shell. The optical properties of the CGS model are better representations of a realistic encapsulated aggregate model than the externally mixed homogeneous spheres, internally mixed homogeneous sphere, and the CS model.

Kahnert et al. (2013) considered that the CGS model consists of BC and sulfate, and the value of $f_{core}$ was optimized to the optical properties of a realistic encapsulated aggregate model. However, we assumed that LA is a mixture of BC and WS, instead of BC and sulfate. WS is defined as a mixture of sulfates, nitrates, organic, water-soluble substances (Hess et al., 1998), and $\omega_0$ of WS is smaller than that of pure sulfate. The optimized values of $f_{core}$ in Kahnert et al. (2013) cannot be applied in this study. Therefore, we optimized $f_{core}$ to the optical properties of the Voronoi aggregate model with BC and WS (Ishimoto et al., 2019). The core of the model is a BC aggregate with a polyhedral Voronoi structure, and the adhering WS shell is created by a simple model of surface tension derived from the artificial surface potential. The refractive index of the BC was obtained from the measurements of Chang and Charalampopoulos (1990). The refractive index of WS depends on the relative humidity and was obtained from the OPAC database (Hess et al., 1998). The optical properties of the Voronoi aggregate model were computed by the finite-difference time-domain method (Ishimoto et al., 2012) and discrete-dipole approximation (DDSCAT version 7.3; https://code.google.com/archive/p/ddscat/ (last access 25 December 2018); Draine and Flatau, 1994). The database of optical properties of the Voronoi aggregate model was created under the following conditions. The volume ratio of shell to core (VR) was 0.0, 2.0, 5.0, 10.0, and 20.0. The volume-equivalent sphere radius was 10 sizes for each VR, and the radius range was from 0.02 to 0.2 µm for VR = 0.0, and from 0.06 to 0.6 µm for VR = 20.0. The relative humidity was 0, 50, 90, and 98 %. The wavelength was 340, 355, 380, 400, 500, 532, 675, 870, 1020, and 1064 nm. These are typical wavelengths of lidar and sky radiometer (Nakajima et al. 2020) measurements. The value of $f_{core}$ was optimized to the optical properties of the Voronoi aggregate model by the following procedure:

(1) $f_{core}$ was changed from 0 to 1 with a step of 0.1.

(2) Optical properties (absorption cross section, $\omega_0$, and $g$) of the CGS model with different values of $f_{core}$ were calculated.

(3) The following objective function was calculated from the optical properties of the CGS and Voronoi aggregate models:

$$\chi(f_{core}) = \sum_{i=1}^{10} \sum_{j=1}^{4} \sum_{k=1}^{5} \left| \frac{\sigma_{abs}^{CGS}(f_{core,,}RH_j,VR_k) - \sigma_{abs}^{Voronoi}(r_i,RH_j,VR_k)}{\sigma_{abs}^{Voronoi}(r_i,RH_j,V_k)} \right| + \left| \frac{\omega_0^{CGS}(f_{core},r_i,RH_j,V_k) - \omega_0^{Voronoi}(r_i,RH_j,VR_k)}{\sigma\omega_0^{Voronoi}(r_i,RH_j,V_k)} \right| +$$
$$\left| \frac{g^{CGS}(f_{core},r_i,RH_j,VR_k) - g^{Voronoi}(r_i,RH_j,VR_k)}{g^{Voronoi}(r_i,RH_j,V_k)} \right|, \quad (A1)$$

where CGS/Voronoi indicate the CGS and Voronoi aggregate models, $\sigma_{abs}$ is the absorption cross section; $r$ is the volume-equivalent sphere radius; $RH$ is relative humidity; and $VR$ is the volume ratio of shell to core. The objective function was calculated for each wavelength and for two particle size ranges, $r < 0.1$ µm and $r \geq 0.1$ µm.

Table A1 shows the objective functions for different values of $f_{core}$. The values of $f_{core} = 0$ and $= 1$ correspond to an internally mixed homogeneous sphere and the CS model, respectively. For $r < 0.1$ µm, the optimized values of $f_{core}$ were 0.8 or 0.9, and the optimized CGS was close to CS. This result is caused by the fact that there are few monomers composing the Voronoi aggregate model when the particle radius is small, and the geometry of the Voronoi aggregate model is close to CS. For $r \geq 0.1$ µm, the $f_{core}$ results were from 0.5 to 0.9. The optimized CGS approached that for internally mixed homogeneous spheres as the wavelength increased. The same wavelength dependency was seen in the results of Kahnert et al. (2013). The optimized $f_{core}$ in Table A1 was used for the calculation of the optical properties of the CGS.

**Data availability.**

The retrieval results of the CALIOP-MODIS retrievals are available on request by contacting the first author of the paper.

**Supplements.**

**Author contributions.**

RK developed the codes of the CALIOP-MODIS retrievals and performed the numerical experiments, and the analysis of the retrieval results. AH, EO, and MF processed the CALIOP and MODIS measurements and ancillary data before the retrievals. HI developed the databases of the Voronoi particle models. TN planed the synergistic remote sensing by CALIOP and MODIS and managed this project. RK prepared the paper with contributions from all co-authors.

**Competing interests.**

The authors declare that they have no conflict of interest.

**Acknowledgments.**

We gratefully acknowledge the uses of PSTAR of the OpenCLASTR (https://ccsr.aori.u-tokyo.ac.jp/~clastr/) and GRASP spheroid-package (https://www.grasp-open.com). We thank the teams involved with CALIPSO, MODIS, and MERRA-2 for their efforts in providing their products. We also thank the AERONET principal investigators and their staff for establishing and maintaining AERONET at the 51 sites used in this investigation.

**Financial support.**

This research has been supported by the Japan Society for the Promotion of Science KAKENHI (grant no. 15H02808).

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

**Table 1. Physical and optical properties of the four aerosol component at relative humidities of 0 and 80 % (0/80 %).**

| Aerosol component | Median radius (µm) | Single-scattering albedo at 532 nm | Asymmetry factor at 532 nm | Lidar ratio at 532 nm (steradians) | Particle depolarization ratio at 532 nm |
|---|---|---|---|---|---|
| Water-soluble | 0.10/0.14 | 0.96/0.98 | 0.50/0.63 | 40/60 | 0.00/0.00 |
| Light-absorbing (Core-grey shell, 30 %*) | 0.10/0.13 | 0.44/0.64 | 0.46/0.59 | 77/92 | 0.00/0.00 |
| Light-absorbing (Core-grey shell, 15 %*) | 0.10/0.14 | 0.58/0.79 | 0.47/0.61 | 61/77 | 0.00/0.00 |
| Light-absorbing (Homogeneous internal mixture, 30 %*) | 0.10/0.13 | 0.46/0.65 | 0.49/0.60 | 88/99 | 0.00/0.00 |
| Light-absorbing (Core-shell 30 %*) | 0.10/0.13 | 0.43/0.61 | 0.43/0.53 | 67/66 | 0.00/0.00 |
| Dust (Voronoi) | 2.00/2.00 | 0.91/0.91 | 0.71/0.71 | 41/41 | 0.49/0.49 |
| Dust (Spheroid) | 2.00/2.00 | 0.92/0.92 | 0.76/0.76 | 51/51 | 0.30/0.30 |
| Sea salt | 2.00/3.99 | 1.00/1.00 | 0.72/0.80 | 13/19 | 0.00/0.00 |

*Volume fraction of black carbon in a particle.

**Table 2. Aerosol components and median radius ($V_{dry}$) values used in the simulations of CALIOP and MODIS observations.**

| Case | Aerosols in the boundary layer | Aerosols in the free atmosphere | $V_{dry}$ of fine (WS, LA)/coarse (DS) particles (μm) |
|---|---|---|---|
| Land Average | External mixture of WS, LA, DS | No aerosols | 0.10/2.00 |
| Land Dust 1 | External mixture of WS, LA, DS | DS | 0.10/2.00 |
| Land Dust 2 | External mixture of WS, LA, DS | DS | 0.10/1.00 |
| Land Dust 3 | External mixture of WS, LA, DS | DS | 0.10/4.00 |
| Land Biomass-Burning 1 | External mixture of WS, LA, DS | External mixture of WS, LA | 0.10/2.00 |
| Land Biomass-Burning 2 | External mixture of WS, LA, DS | External mixture of WS, LA | 0.07/2.00 |
| Land Biomass-Burning 3 | External mixture of WS, LA, DS | External mixture of WS, LA | 0.15/2.00 |
| Land Polluted Dust | External mixture of WS, LA, DS | External mixture of WS, LA, DS | 0.10/2.00 |
| Ocean Clean | External mixture of WS, SS | No aerosols | 0.10/2.00 |
| Ocean Dust 1 | External mixture of WS, SS | DS | 0.10/2.00 |
| Ocean Dust 2 | External mixture of WS, SS | DS | 0.10/1.00 |
| Ocean Dust 3 | External mixture of WS, SS | DS | 0.10/4.00 |
| Ocean Biomass-Burning 1 | External mixture of WS, SS | External mixture of WS, LA | 0.10/2.00 |
| Ocean Biomass-Burning 2 | External mixture of WS, SS | External mixture of WS, LA | 0.07/2.00 |
| Ocean Biomass-Burning 3 | External mixture of WS, SS | External mixture of WS, LA | 0.15/2.00 |
| Ocean Polluted Dust | External mixture of WS, SS | External mixture of WS, LA, DS | 0.10/2.00 |


**Table 3. Means and standard deviations of differences of columnar properties between retrievals and simulations.**

| Parameter | Aerosol optical depth at 532 nm | Land | | Ocean | |
|---|---|---|---|---|---|
| | | 532 nm | 1064 nm | 532 nm | 1064 nm |
| Aerosol optical depth[a] | <0.3 | −2 ±10 % | 0 ± 14 % | −15 ± 25 % | −10 ± 10 % |
| | ≥0.3 | −3 ± 11 % | −2 ± 0.13 % | 10 ± 13 % | 6 ± 11 % |
| Aerosol optical depth of water-soluble particles[a] | <0.3 | −9 ± 15 % | | −31 ± 39 % | |
| | ≥0.3 | −4 ± 12 % | | 5 ± 23 % | |
| Aerosol optical depth of light-absorbing particles[a] | < 0.3 | 114 ± 131 % | | 27 ± 86 % | |
| | ≥0.3 | 24 ± 99 % | | 78 ± 85 % | |
| Aerosol optical depth of dust[a] | <0.3 | 15 ± 167 % | | −17 ± 11 % | |
| | ≥0.3 | 5 ± 153 % | | −9 ± 8 % | |
| Aerosol optical depth of sea salt[a] | <0.3 | | | 41 ± 50 % | |
| | ≥0.3 | | | −2 ± 30 % | |
| Single-scattering albedo[b] | <0.3 | −0.02 ± 0.05 | −0.01 ± 0.08 | −0.01 ± 0.04 | 0.01 ± 0.06 |
| | ≥0.3 | −0.01 ± 0.03 | −0.01 ± 0.04 | −0.03 ± 0.04 | −0.03 ± 0.05 |
| Asymmetry factor[b] | <0.3 | 0.02 ± 0.03 | 0.00 ± 0.03 | 0.03 ± 0.04 | 0.04 ± 0.05 |
| | ≥0.3 | 0.01 ± 0.02 | 0.00 ± 0.02 | 0.00 ± 0.02 | −0.02 ± 0.04 |
| Dry median radius of fine particles[a] | <0.3 | 9 ± 10 % | | 4 ± 11 % | |
| | ≥0.3 | 4 ± 8 % | | 3 ± 8 % | |
| Dry median radius of coarse particles[a] | <0.3 | 8 ± 27 % | | 11 ± 39 % | |
| | ≥0.3 | 6 ± 18 % | | 6 ± 15 % | |

[a]Differences are calculated by $100 \times (Retrieval - Simulation) / Simulation$.

[b]Differences are calculated by $(Retrieval - Simulation)$


**Table 4. Means and standard deviations of differences of vertically resolved properties between retrievals and simulations.**

| Parameter at 532 nm | Aerosol optical depth at 532 nm | Land | Ocean |
|---|---|---|---|
| Extinction coefficient[a] | <0.3 | −4 ± 19 % | −17 ± 35 % |
| | ≥0.3 | −4 ± 16 % | 9 ± 26 % |
| Extinction coefficient of water-soluble[a] | <0.3 | −15 ± 30 % | −35 ± 54 % |
| | ≥0.3 | −7 ± 22 % | 6 ± 42 % |
| Extinction coefficient of light-absorbing[a] | <0.3 | 185 ± 366 % | 11 ± 84 % |
| | ≥0.3 | 30 ± 172 % | 54 ± 95 % |
| Extinction coefficient of dust[a] | <0.3 | −6 ± 150 % | −18 ± 13 % |
| | ≥0.3 | −5 ± 128 % | −10 ± 10 % |
| Extinction coefficient of sea salt[a] | <0.3 | | 37 ± 46 % |
| | ≥0.3 | | −2 ± 34 % |
| Single-scattering albedo[b] | <0.3 | −0.03 ± 0.09 | −0.01 ± 0.05 |
| | ≥0.3 | −0.01 ± 0.05 | −0.03 ± 0.06 |
| Asymmetry factor[b] | <0.3 | 0.02 ± 0.03 | 0.04 ± 0.06 |
| | ≥0.3 | 0.01 ± 0.02 | 0.00 ± 0.03 |

[a]Differences are calculated by $100 \times (Retrieval - Simulation)/Simulation$.

[b]Differences are calculated by $(Retrieval - Simulation)$.


**Table 5. Means and standard deviations of deviations of the retrieval results using different particle models compared with the retrieval result using the Voronoi model for dust and the core-grey shell 30 %\* model for light-absorbing particles.**

| Parameter | Spheroid for dust | Core-grey shell 15%* for light-absorbing | Homogeneous mixture 30%* for light-absorbing particles | Core-shell 30%* for light-absorbing particles |
|---|---|---|---|---|
| Aerosol optical depth of water-soluble particles at 532 nm | $-0.005 \pm 0.019$ | $-0.004 \pm 0.012$ | $0.001 \pm 0.008$ | $-0.005 \pm 0.013$ |
| Aerosol optical depth of light-absorbing particles at 532 nm | $-0.004 \pm 0.009$ | $-0.006 \pm 0.011$ | $0.002 \pm 0.006$ | $-0.008 \pm 0.013$ |
| Aerosol optical depth of dust at 532 nm | $0.015 \pm 0.038$ | $0.000 \pm 0.005$ | $0.000 \pm 0.003$ | $0.000 \pm 0.004$ |
| Aerosol optical depth of sea salt at 532 nm | $-0.002 \pm 0.005$ | $0.001 \pm 0.005$ | $0.000 \pm 0.004$ | $0.001 \pm 0.003$ |
| Dry median radius of fine particles ($\mu$m) | $-0.002 \pm 0.004$ | $0.001 \pm 0.003$ | $0.000 \pm 0.002$ | $0.001 \pm 0.004$ |
| Dry median radius of coarse particles (µm) | $-0.071 \pm 0.109$ | $0.029 \pm 0.096$ | $0.005 \pm 0.060$ | $0.016 \pm 0.089$ |

*Volume fraction of black carbon in a particle.

**Table A1. Objective function (Eq. (A1)) for different volume fractions of BC core ($f_{core}$) in a particle, volume-equivalent sphere radius ranges, and wavelengths. Bold underlined text indicates the minimum value of the objective function in each row.**

| Radius (μm) | Wavelength (μm) | $f_{core}$ | | | | | | | | | | |
|---|---|---|---|---|---|---|---|---|---|---|---|---|
| | | 0.0 | 0.1 | 0.2 | 0.3 | 0.4 | 0.5 | 0.6 | 0.7 | 0.8 | 0.9 | 1.0 |
| <0.1 | 0.340 | 0.038 | 0.038 | 0.038 | 0.038 | 0.037 | 0.035 | 0.032 | 0.027 | **0.021** | 0.022 | 0.037 |
| | 0.355 | 0.039 | 0.039 | 0.038 | 0.037 | 0.036 | 0.033 | 0.029 | 0.024 | **0.017** | 0.019 | 0.038 |
| | 0.380 | 0.047 | 0.047 | 0.047 | 0.045 | 0.043 | 0.039 | 0.032 | 0.024 | **0.014** | 0.017 | 0.040 |
| | 0.400 | 0.059 | 0.059 | 0.058 | 0.057 | 0.053 | 0.048 | 0.040 | 0.029 | 0.017 | **0.015** | 0.039 |
| | 0.500 | 0.064 | 0.064 | 0.064 | 0.062 | 0.058 | 0.053 | 0.044 | 0.032 | 0.021 | **0.016** | 0.035 |
| | 0.532 | 0.064 | 0.064 | 0.063 | 0.062 | 0.058 | 0.053 | 0.044 | 0.033 | 0.022 | **0.017** | 0.033 |
| | 0.675 | 0.053 | 0.053 | 0.053 | 0.051 | 0.048 | 0.043 | 0.036 | 0.028 | 0.021 | **0.020** | 0.032 |
| | 0.870 | 0.045 | 0.045 | 0.044 | 0.043 | 0.041 | 0.038 | 0.034 | 0.031 | **0.029** | 0.032 | 0.042 |
| | 1.020 | 0.050 | 0.050 | 0.050 | 0.049 | 0.047 | 0.045 | 0.042 | 0.041 | **0.040** | 0.045 | 0.055 |
| | 1.064 | 0.053 | 0.053 | 0.053 | 0.052 | 0.051 | 0.049 | 0.047 | 0.046 | **0.045** | 0.051 | 0.062 |
| ≥0.1 | 0.340 | 0.142 | 0.142 | 0.142 | 0.140 | 0.136 | 0.128 | 0.114 | 0.092 | 0.060 | **0.049** | 0.158 |
| | 0.355 | 0.124 | 0.124 | 0.124 | 0.122 | 0.118 | 0.111 | 0.098 | 0.077 | **0.050** | 0.052 | 0.149 |
| | 0.380 | 0.092 | 0.092 | 0.092 | 0.091 | 0.087 | 0.081 | 0.070 | 0.054 | **0.038** | 0.060 | 0.137 |
| | 0.400 | 0.072 | 0.072 | 0.072 | 0.071 | 0.068 | 0.062 | 0.053 | 0.041 | **0.034** | 0.062 | 0.126 |
| | 0.500 | 0.039 | 0.039 | 0.039 | 0.038 | 0.035 | 0.032 | 0.027 | **0.027** | 0.037 | 0.061 | 0.099 |
| | 0.532 | 0.035 | 0.035 | 0.035 | 0.034 | 0.032 | 0.029 | **0.026** | 0.027 | 0.037 | 0.057 | 0.090 |
| | 0.675 | 0.035 | 0.035 | 0.035 | 0.034 | 0.032 | 0.030 | **0.029** | 0.030 | 0.034 | 0.046 | 0.067 |
| | 0.870 | 0.041 | 0.041 | 0.041 | 0.040 | 0.039 | 0.038 | **0.037** | 0.037 | 0.041 | 0.051 | 0.067 |
| | 1.020 | 0.048 | 0.048 | 0.048 | 0.048 | 0.047 | **0.047** | 0.047 | 0.049 | 0.055 | 0.067 | 0.086 |
| | 1.064 | 0.050 | 0.050 | 0.050 | 0.050 | 0.050 | **0.049** | 0.050 | 0.052 | 0.058 | 0.071 | 0.092 |

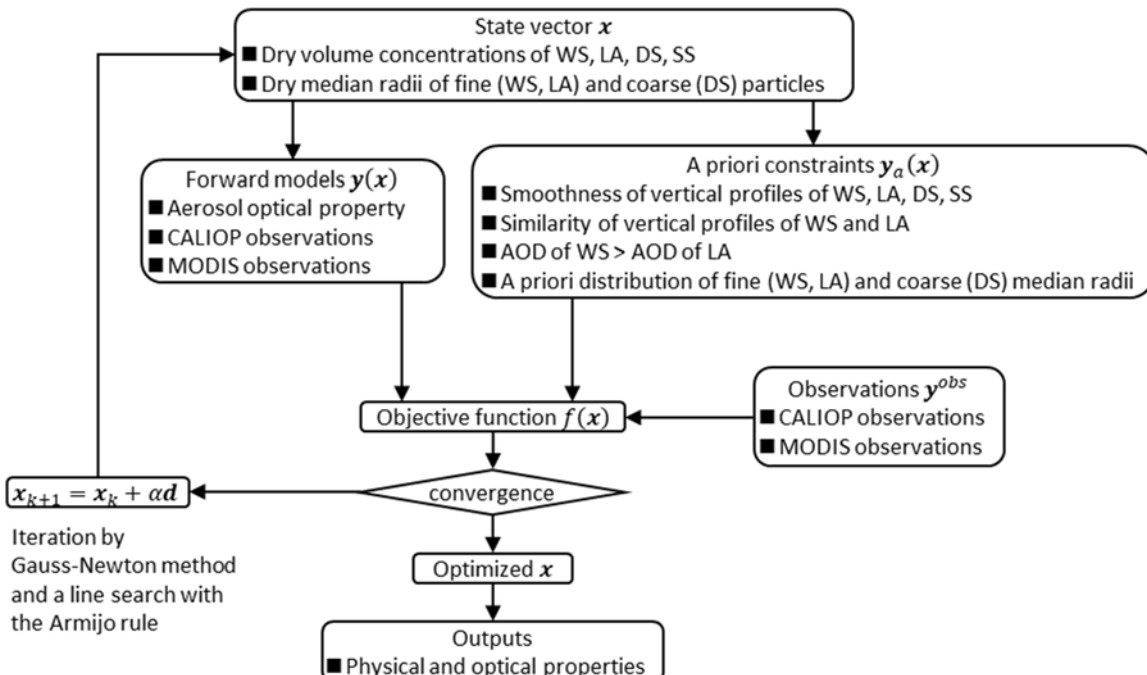

**Figure 1. Schematic diagram of the retrieval procedures.**

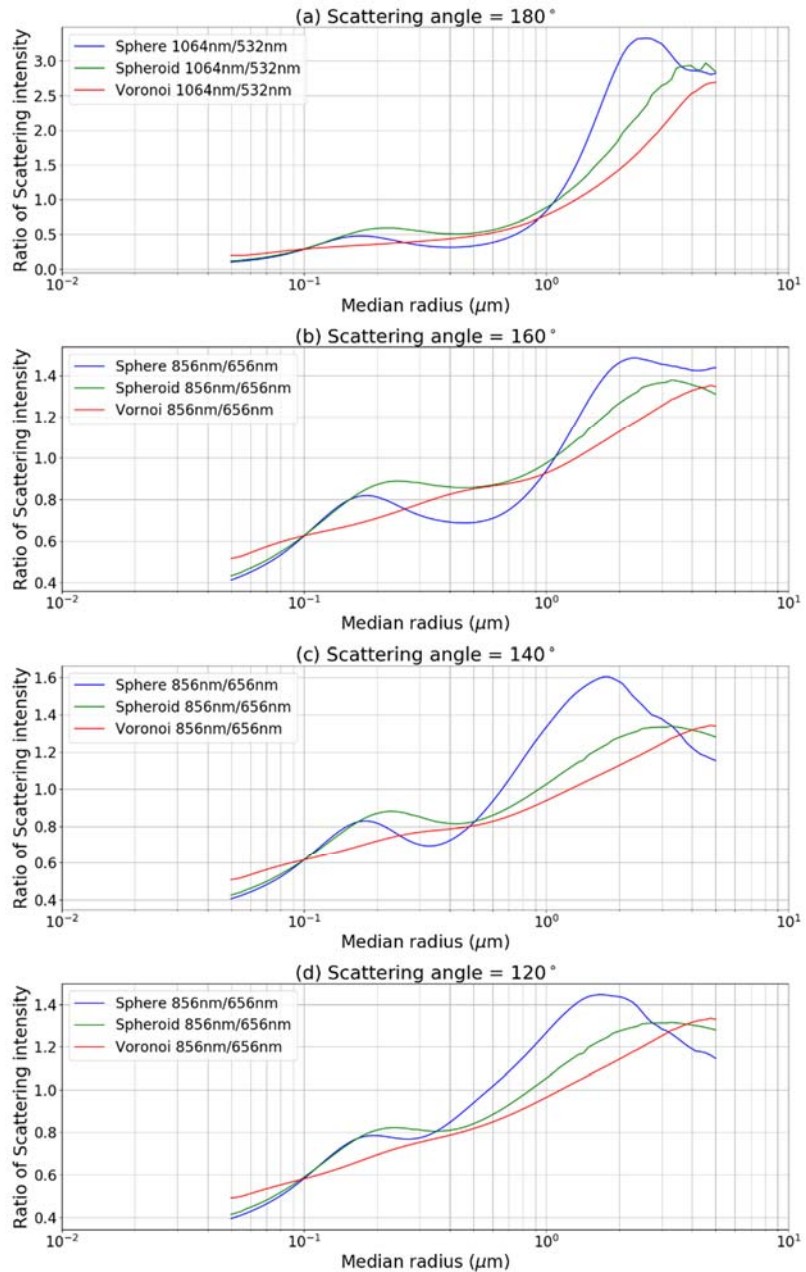

**Figure 2. Relation between median radius and the ratio of scattering intensity at different wavelengths for (a) CALIOP and (b, c, and d) MODIS observations. Blue, green, and red colours indicate sphere, spheroid, and Voronoi particle models, respectively.**


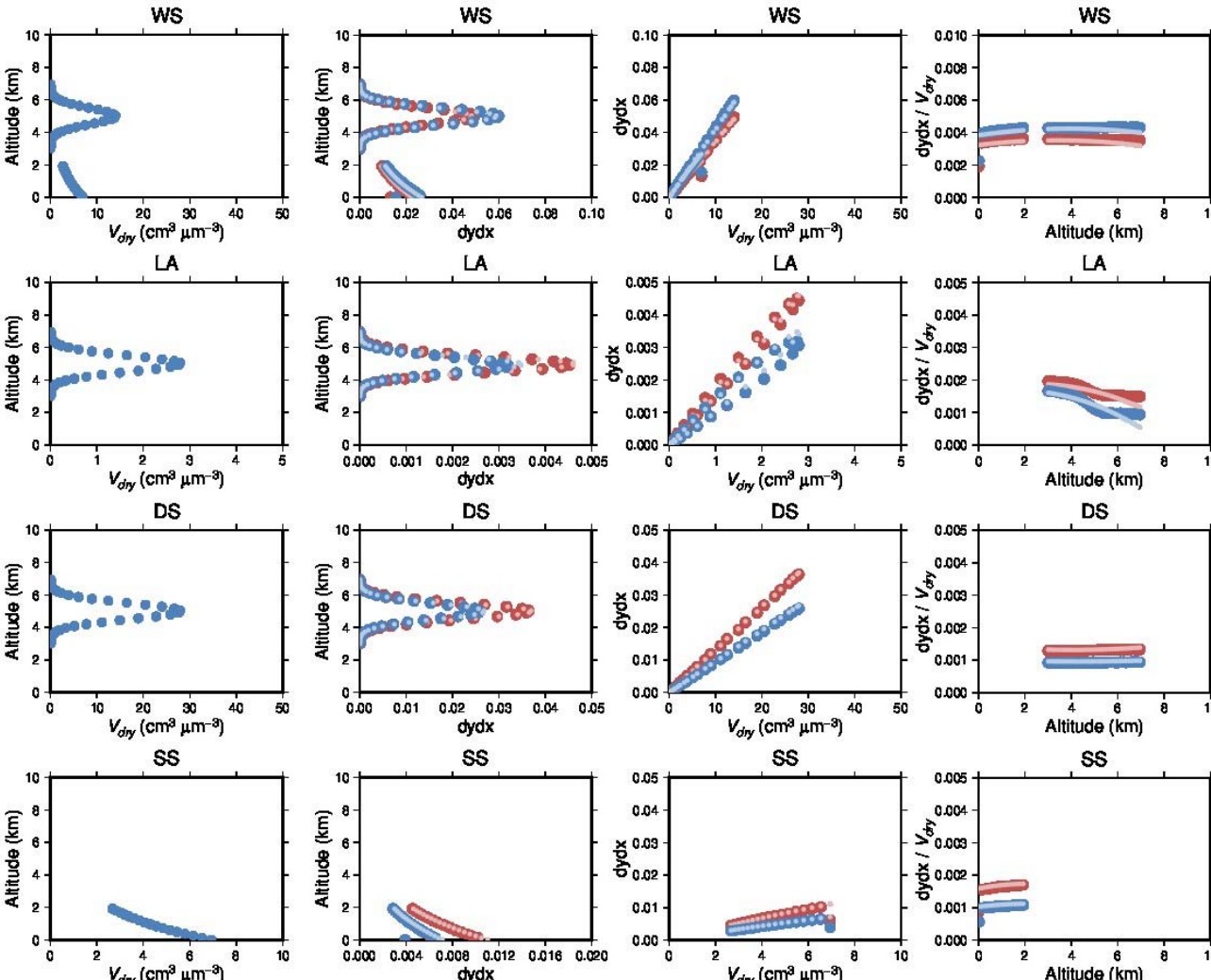

**Figure 3.** Approximation of the numerical derivatives of MODIS radiances for $V_{dry}$ of WS (first row), LA (second row), DS (third row), and SS (fourth row). The first column shows vertical profiles of $V_{dry}$; the second column shows vertical profiles of the numerical derivatives (dydx); the third column shows the dependency of dydx on $V_{dry}$; and the fourth column shows the dependency of dydx/$V_{dry}$ on altitude. Blue and red colours indicate dydx at MODIS bands 1 and 2, respectively. Dark and light colours indicate the reference values and the approximated calculations, respectively.

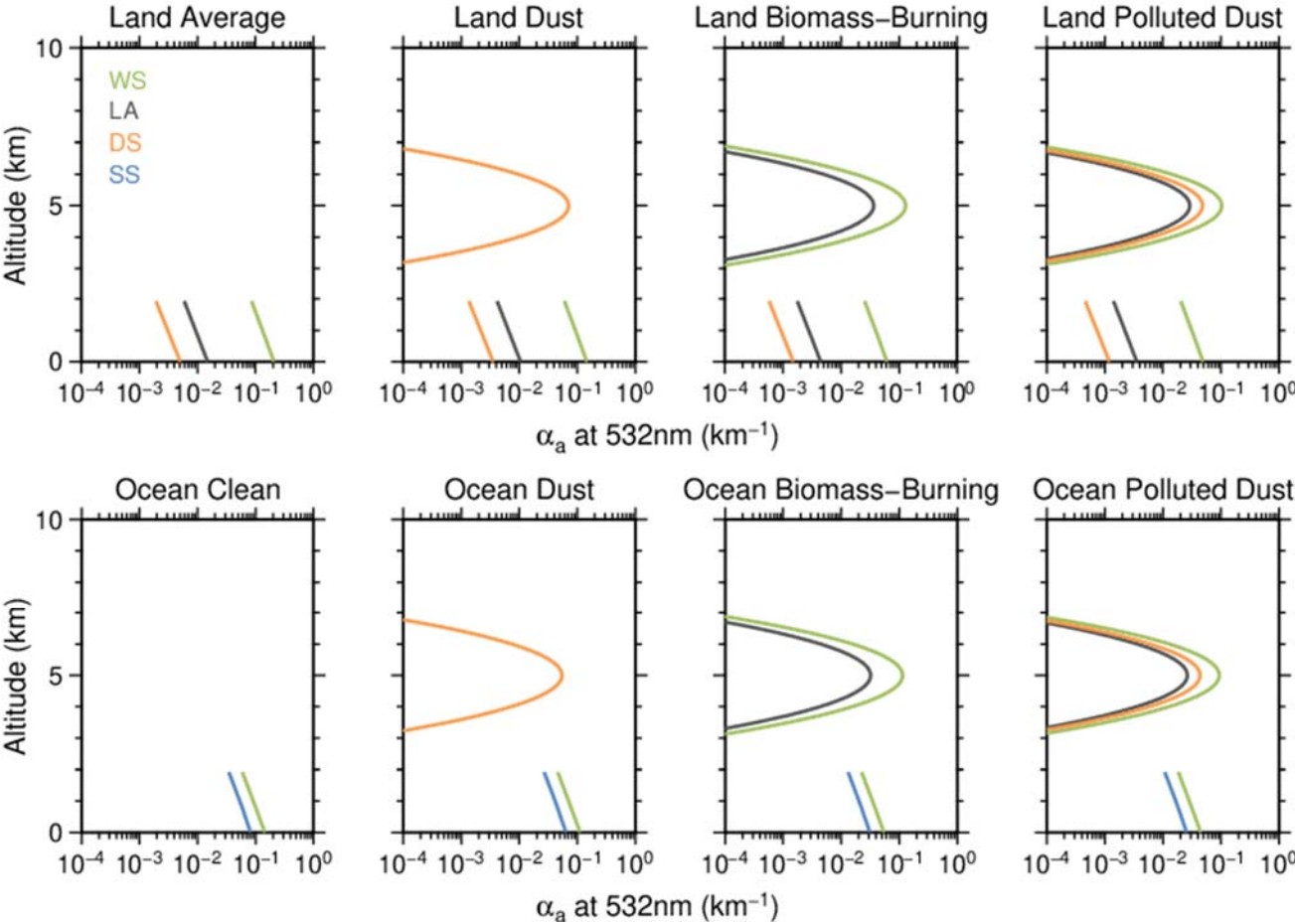

Figure 4. Vertical profiles of $\alpha_a$ of WS (green), LA (black), DS (orange), and SS (blue) used in the simulations of the clean, average, dust, biomass-burning, and polluted dust cases over land and ocean surfaces. Total $\tau_a$ in all panels is 0.3 at 532 nm.

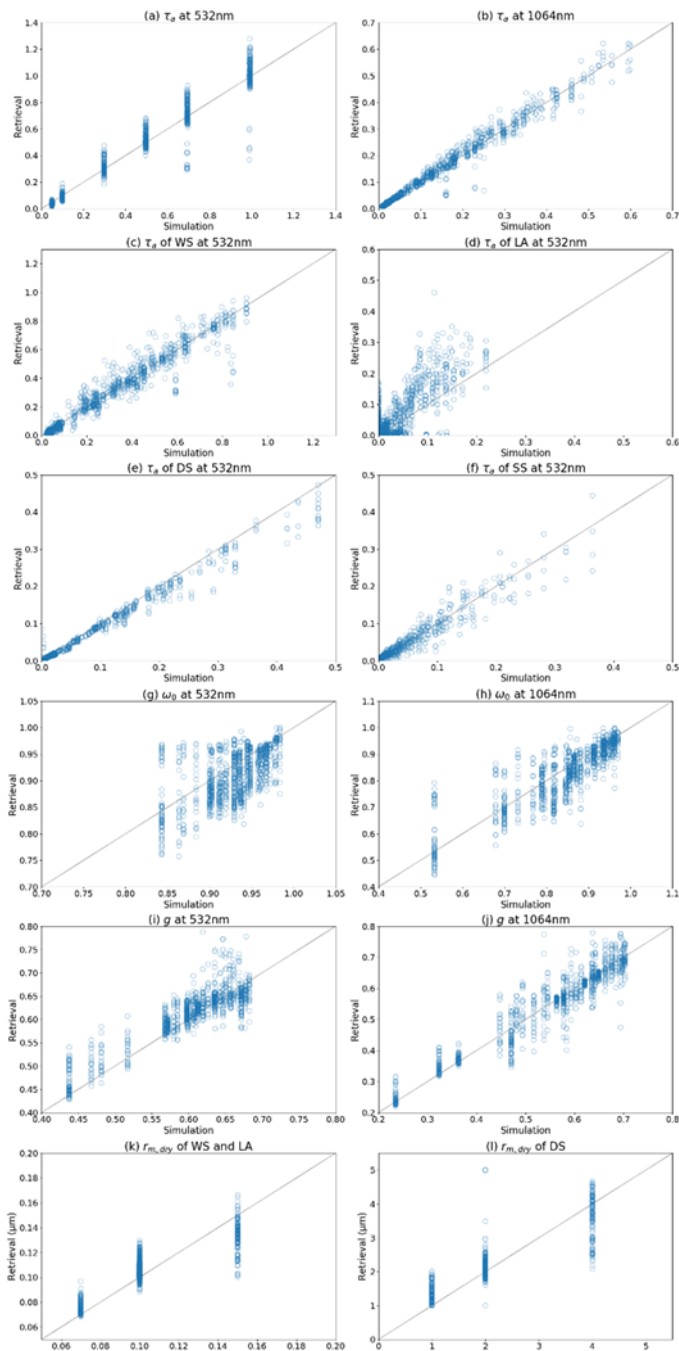

**Figure 5.** Scatter plots of simulated and retrieved columnar properties: $\tau_a$ at (a) 532 and (b) 1064 nm; $\tau_a$ at 532 nm of (c) WS, (d) LA, (e) DS, and (f) SS; $\omega_0$ at (g) 532 and (h) 1064 nm; $g$ at (i) 532 and (j) 1064 nm; $r_{m,dry}$ of (k) fine (WS and LA) and (l) coarse (DS) particles.

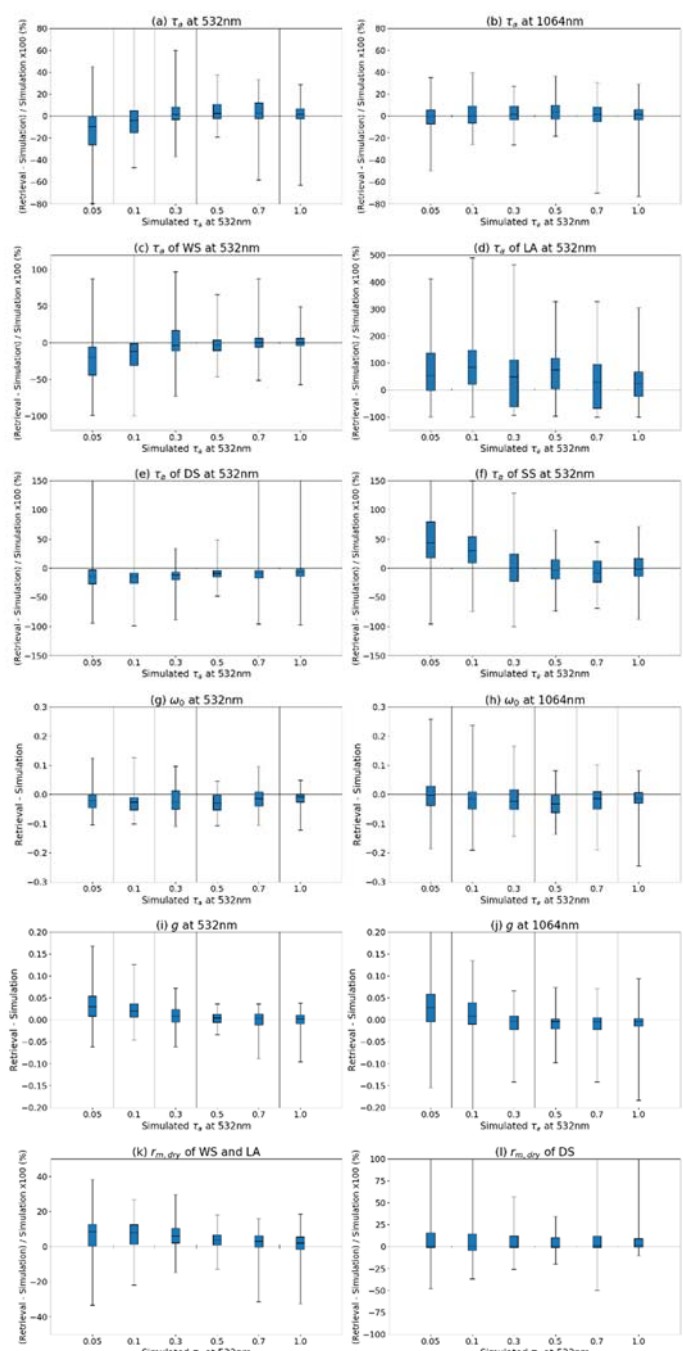

**Figure 6. Box and whisker plots for relative or absolute differences of columnar properties between retrievals and simulations. The box extends from the first quartile to the third quartile of the data, with a line at the median. The whiskers extend from the box to 1.5 × inter-quartile range. The column properties are $\tau_a$ at (a) 532 and (b) 1064 nm; $\tau_a$ at 532 nm of (c) WS, (d) LA, (e) DS, and (f) SS; $\omega_0$ at (g) 532 and (h) 1064 nm; $g$ at (i) 532 and (j) 1064 nm; and $r_{m,dry}$ of (k) fine (WS and LA) and (l) coarse (DS) particles.**


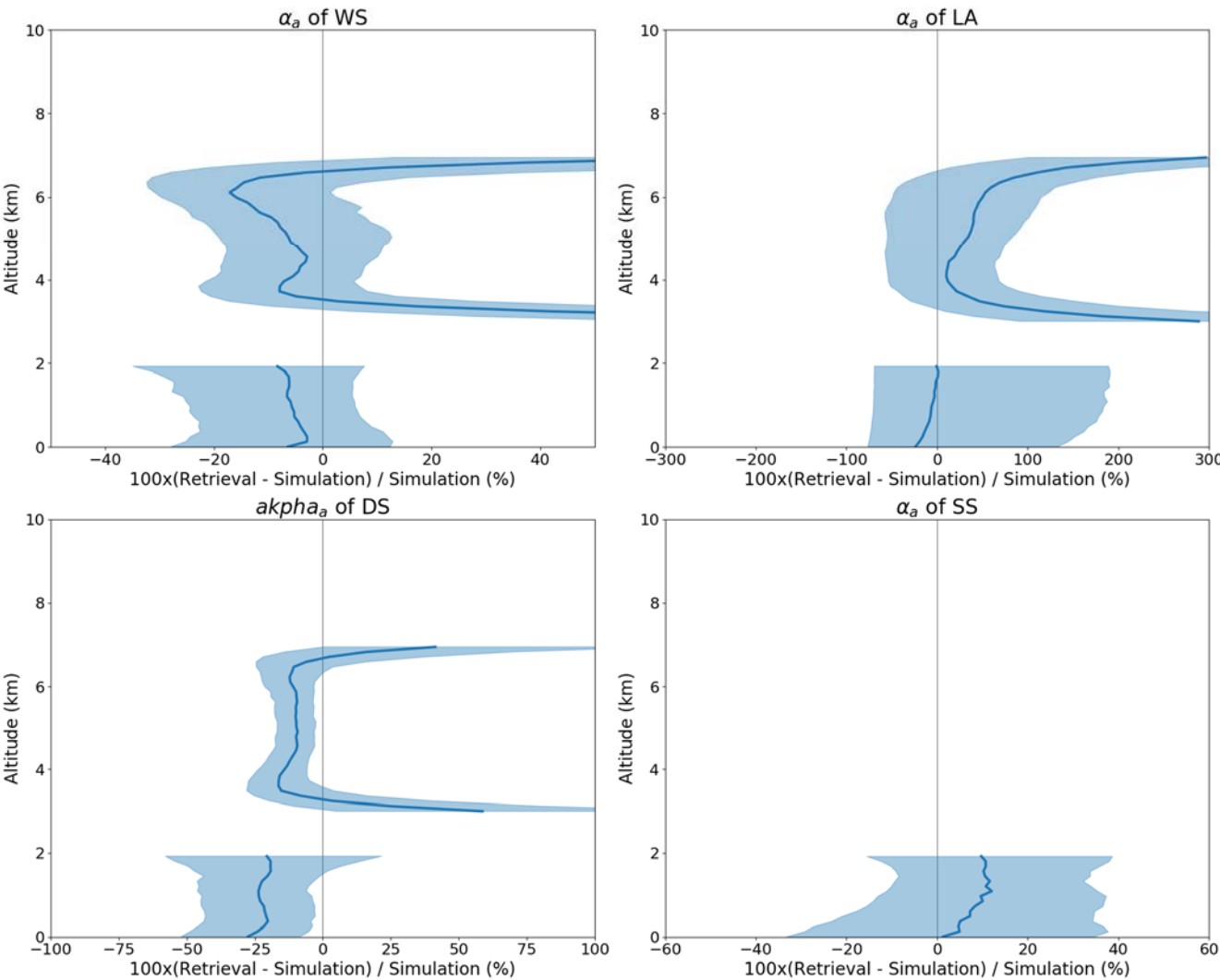

**Figure 7. Relative differences of $\alpha_a$ at 532 nm for (a) WS, (b) LA, (c) DS, and (d) SS between retrievals and simulations. The shading indicates the areas between the first and third quartiles of the data, and the thick lines indicate median values.**

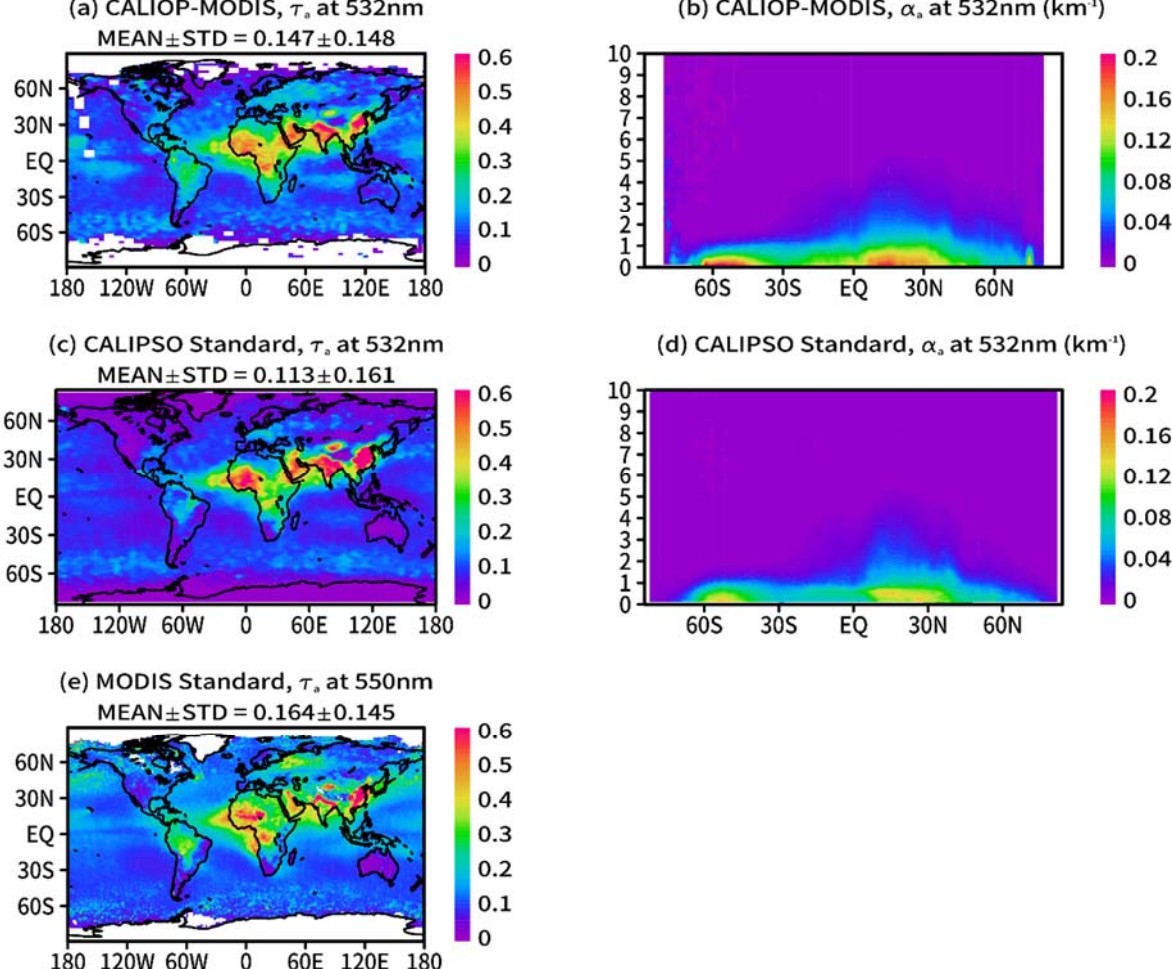

**Figure 8. Annual means of $\tau_a$ and $\alpha_a$ in 2010. The left column shows horizontal distributions of $\tau_a$, and the right column shows zonal means of $\alpha_a$ for the (a, b) CALIOP-MODIS retrieval, (c, d) CALIPSO standard product, and (e) MODIS standard product. At the top of the left panels, MEAN±STD indicates the global mean and its standard deviation.**

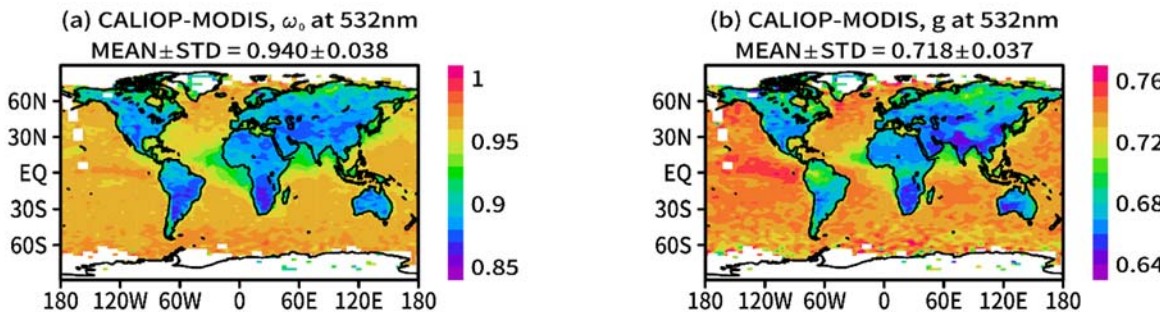

**Figure 9. Horizontal distributions of the annual means of (a) $\omega_0$ and (b) $g$ in 2010 in the CALIOP-MODIS retrieval. At the top of each panel, MEAN±STD indicates the global mean and its standard deviation.**

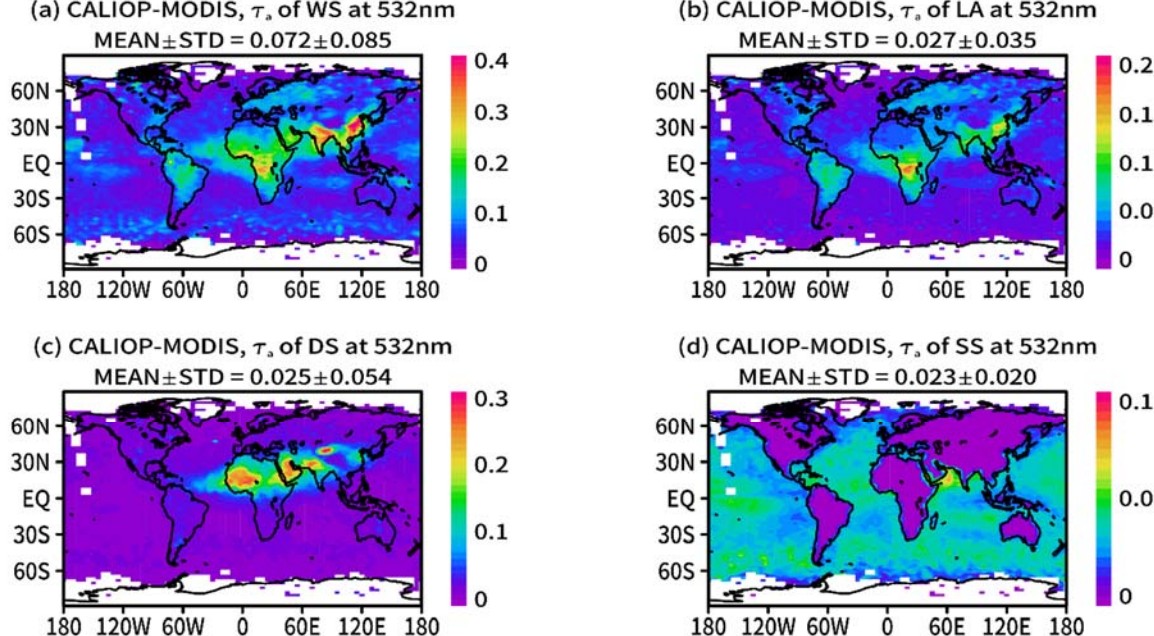


**Figure 10. Horizontal distributions of the annual means of $\tau_a$ of (a) WS, (b) LA, (c) DS, and (d) SS in 2010 in the CALIOP-MODIS retrieval. At the top of each panel, MEAN±STD indicates the global mean and its standard deviation.**

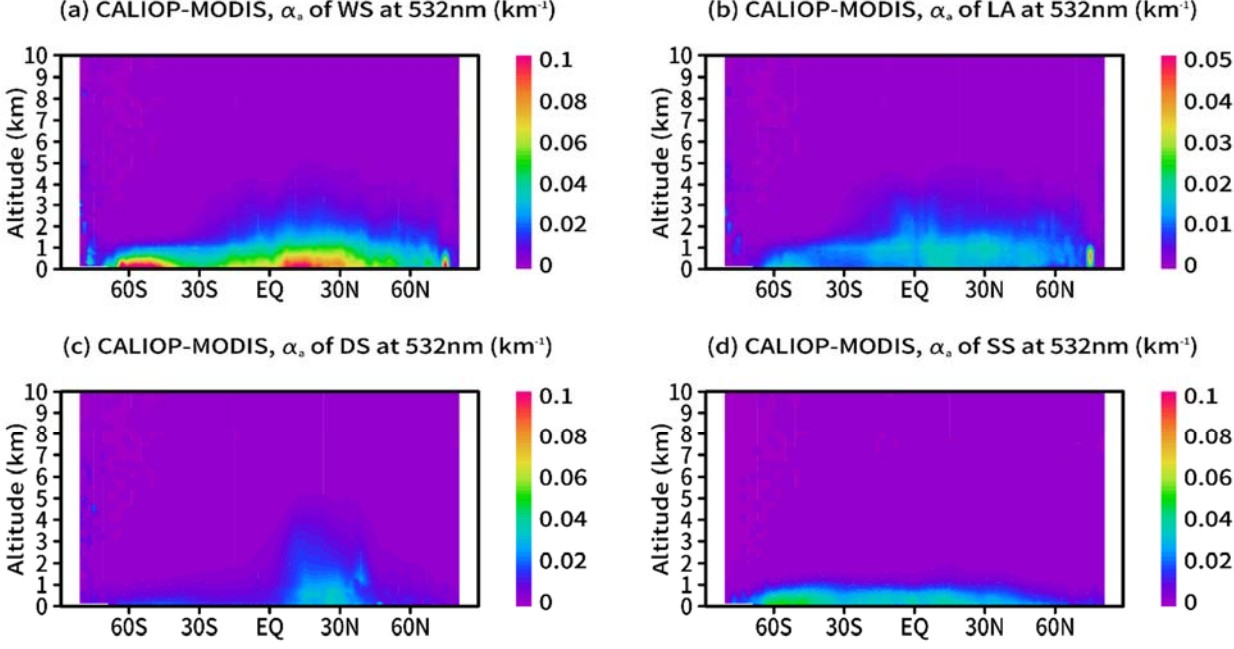


**Figure 11. Zonal means of $\alpha_a$ of (a) WS, (b) LA, (c) DS, and (d) SS in 2010 in the CALIOP-MODIS retrieval.**

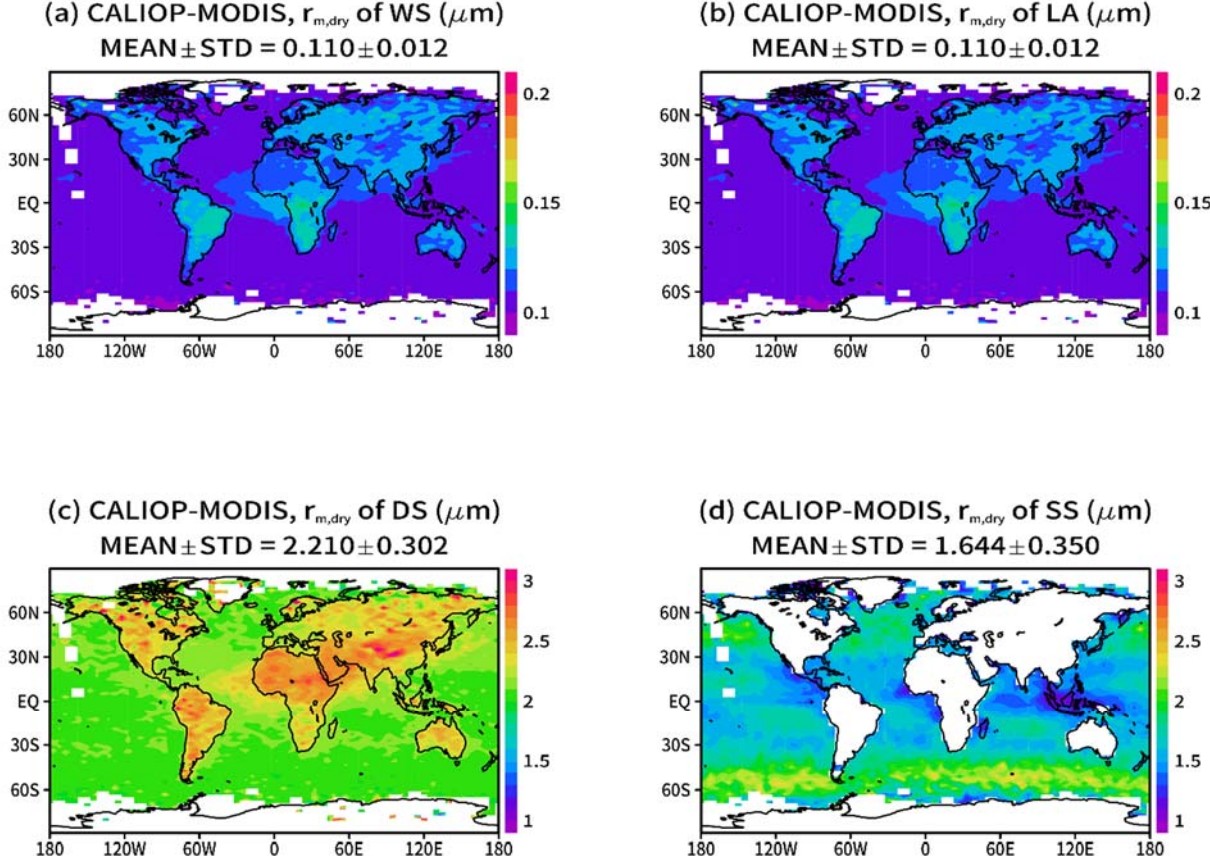

**Figure 12. Horizontal distributions of the annual means of $r_{m,dry}$ of (a) WS, (b) LA, (c) DS, and (d) SS in 2010 in the CALIOP-MODIS retrieval. At the top of each panel, MEAN$\pm$STD indicates the global mean and its standard deviation.**

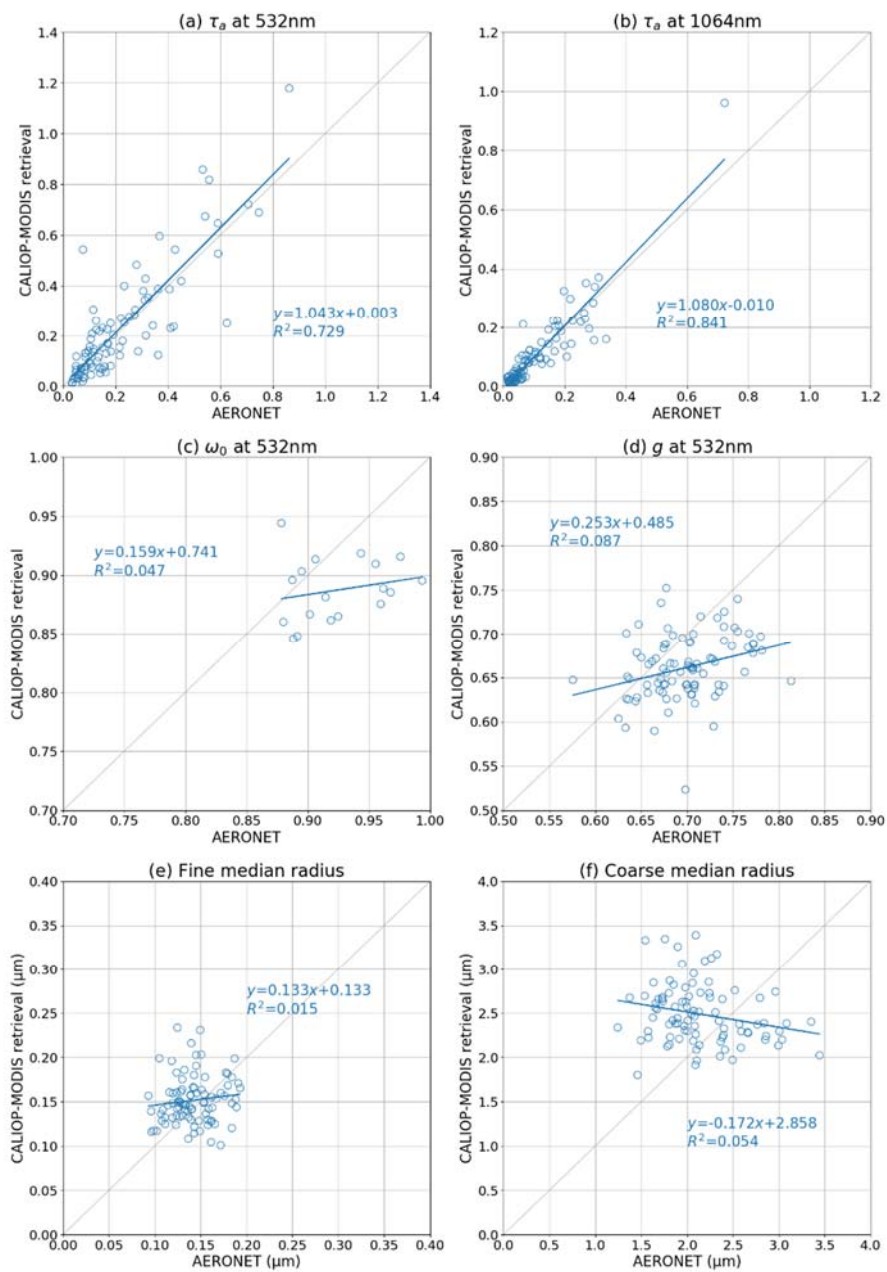

**Figure 13. Comparisons of the columnar properties between the AERONET products and CALIOP-MODIS retrieval: $\tau_a$ at (a) 532 nm and (b) 1064 nm; (c) $\omega_0$ at 532 nm; (d) $g$ at 532 nm; (e) fine median radius, and (f) coarse median radius. The linear regression results are shown as equations in the form y = ax + b, and $R^2$ is the coefficient of determination.**


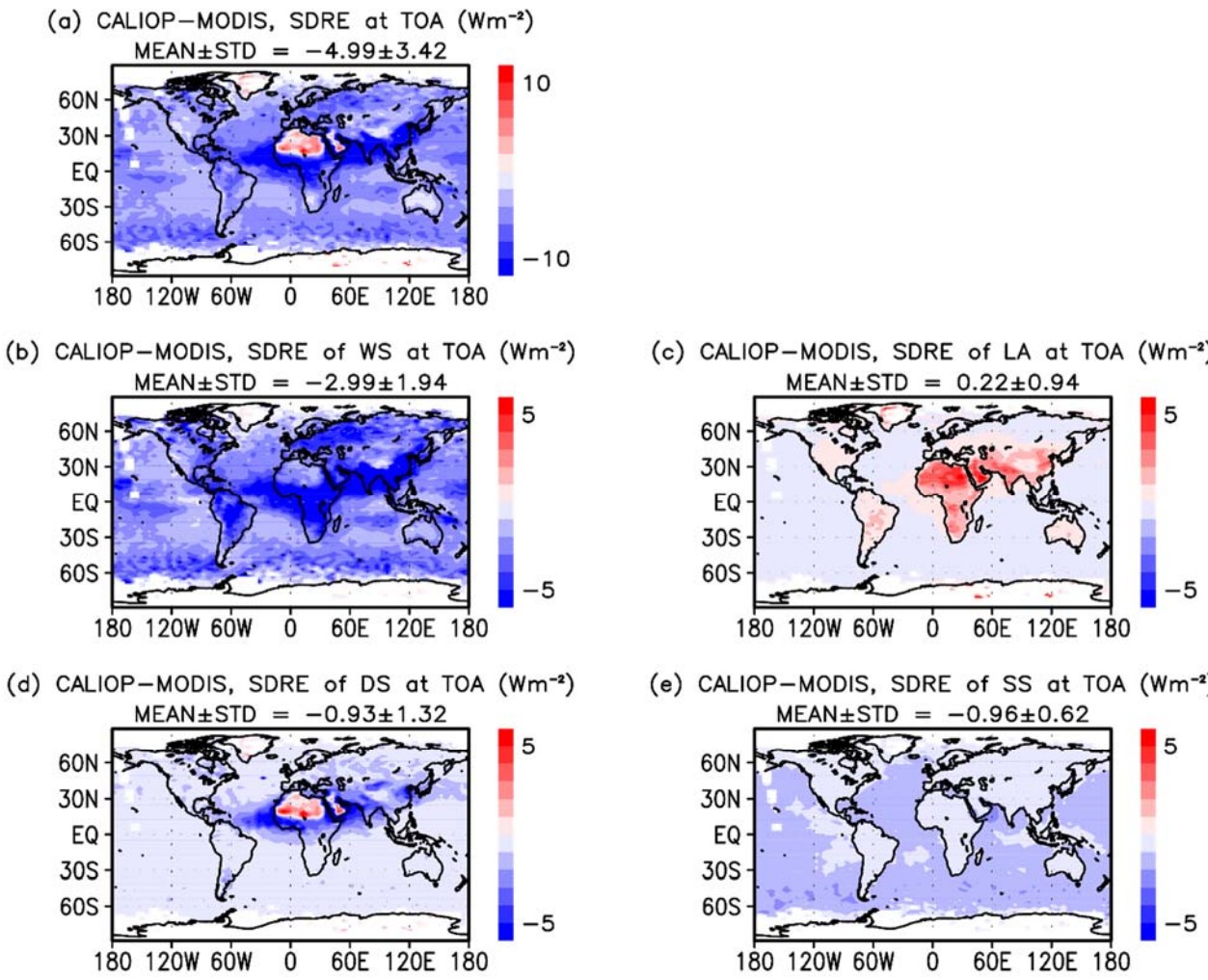

**Figure 14. Horizontal distributions of the annual means of the SDRE of (a) total aerosols, (b) WS, (c) LA, (d) DS, and (e) SS (e) at top of the atmosphere (TOA) in 2010. At the top of each panel, MEAN±STD indicates the global mean and its standard deviation.**


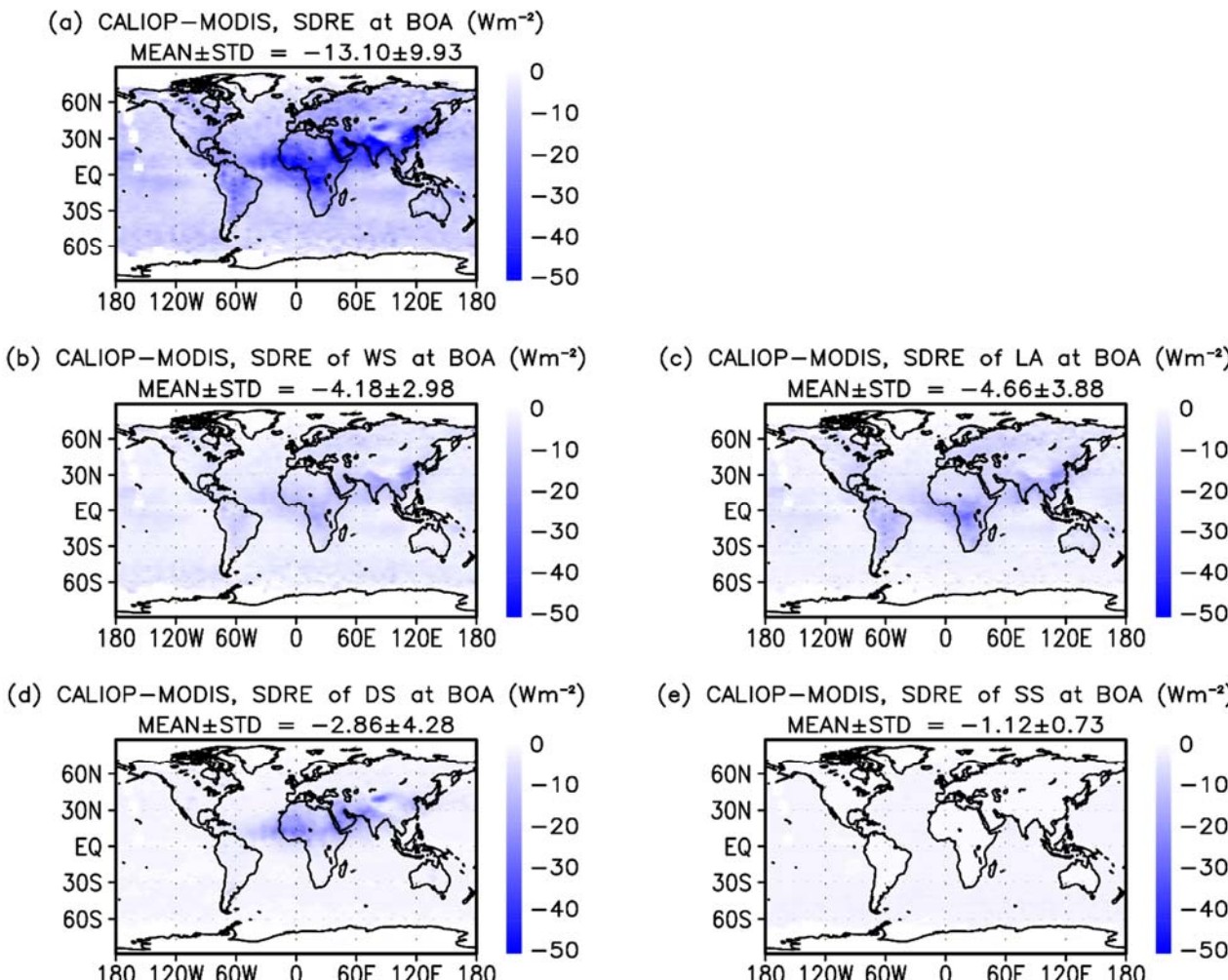

**Figure 15. Horizontal distributions of the annual means of the SDRE values of (a) total aerosols, (b) WS, (c) LA, (d) DS, and (e) SS at the bottom of the atmosphere (BOA) in 2010. At the top of each panel, MEAN±STD indicates the global mean and its standard deviation.**


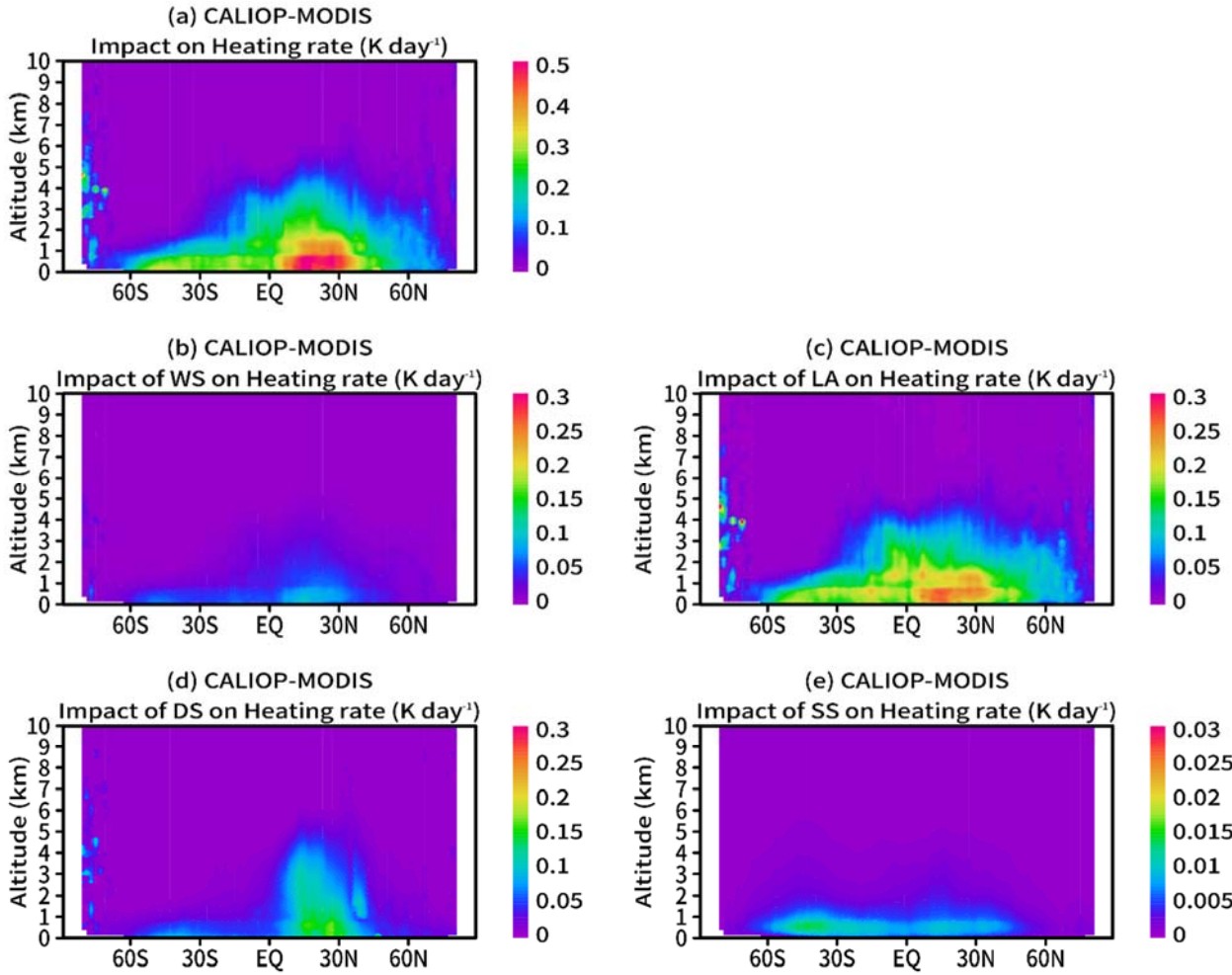

**Figure 16. Annual means of impacts of (a) total aerosols, (b) WS, (c) LA, (d) DS, and (e) SS on heating rates in 2010.**