# Peer review of "Global 3-D distribution of aerosol composition by synergistic use of CALIOP and MODIS observations"

_Atmospheric Measurement Techniques, 2023_

## Referee Comment (RC2)

A nice structured work, developed by the utilization of CALIOP and MODIS retrievals for the establishment of a global aerosol-speciated 3D distribution. Typical aerosol properties are derived and collocated against ground-based stations (AERONET). Finally, SDRF values (under clear sky conditions) are retrieved and compared against results in previous studies for the estimation of aerosol induced perturbations on the Earth-Atmosphere radiation budget.

**1 Introduction**

I think the revised V4 types of CALIPSO and some weaknesses of CALIOP and MODIS retrievals - not only the limited wavelength information and the strong surface reflectance, respectively - should be mentioned (these preferences would probably have a reasonable contribution to the uncertainty in some CALIOP-MODIS retrievals).

In Figure 8 the different strong aerosol sources (e.g. dust source in the region of Bodélé) are not visible. For example, a well-known problem of CALIOP-CALIPSO retrievals is the sufficient underestimation of AOD over strong aerosol sources, an inadequacy strongly related to the presence opaque layers completely attenuating the laser beam. Probably a colorbar with a lower AOD limit (less than 0.8) or with modified bins or just a different colorbar could help with the visualization of this result. If a filter is applied for the smoothness of the colors on the map, this filter maybe contaminates the AOD over the sources especially if the surrounding regions have substantially lower AOD.

In Figure 9 an aerosol-speciated distribution is not clear. It's like having 2 groups of SSA values (land-ocean). A narrower colorbar (starting e.g. from 0.8) could help with the distinguishing of some areas. For example, over the Northern and the Central Africa a lower and a higher SSA value should be visible (dust and more absorbing particles-like smoke from biomass burning-respectively). The same problem is visible for AF.

In Figure 12 it's not clear for me some hotspots of coarse DS particles over the Norway and Sweden

In Figure 13 AOD shows a good agreement with AERONET, but the other parameters rather deviate. In comparison with Figure 9 maybe the results for the other properties need further investigation, since these parameters are also used for the radiative simulations and furthermore for the heating rate.

---

## Referee Comment (RC4)

**Review report amt-2023-59 manuscript**

The current study deals with the development of a joint retrieval algorithm utilizing CALIOP and MODIS observations. In the algorithm, four aerosol species (water-soluble, light-absorbing, dust and sea-salt) are assumed and for each one of them optical and microphysical properties are derived for 2010. The obtained products are compared against those provided by the official CALIOP and MODIS products as well as against AERONET observations. Moreover, the authors have performed an analysis of the aerosol-induced direct radiative effects. Overall, it is an interesting study which has a lot of potential. However, there are several parts in the manuscript which must be improved thus helping the reader to understand the approach and the scope of each step of the applied methodology (Section 3.1). Another weak point of the study is the poor interpretation of the results presented in Section 5. To be more specific, the discussion should not be limited only in the description of the plots but it should be associated with a physical interpretation. I would also suggest the authors to revise the English writing style throughout the text. As a conclusion, I believe that the submitted paper fits to the AMT purposes but it is needed a major revision before being accepted for publication. I hope that my comments, which are listed below, will help the authors to improve their work.

1. **Line 35:** Aerosols interact also with the longwave radiation.
2. It is better to use the term radiative effect rather than radiative forcing since the latter one is related to the induced perturbations of the radiation budget attributed to anthropogenic activities.
3. **Lines 46-48:** Since you are mentioning the Aeronet optical properties derived by almucantar retrievals you must cite the relevant publications from Oleg Dubovik.
4. **Lines 51-53:** Can you be more specific? Which imagers?
5. **Lines 55-58:** Why is important to mention the aerosol subtypes in the CALIOP Version 3 data since they are not used in the current study?
6. **Lines 70-81:** I suggest to describe explicitly which are the advantages and disadvantages of the CALIOP and MODIS instruments and how complement each other. The current description is poor and vague. It is needed a better description supported by findings of previous studies. What do you mean *"To observe the global three-dimensional distribution of the aerosol composition, we have developed two aerosol composition retrieval methods that use the observations of CALIOP and the Moderate Resolution Imaging Spectroradiometer (MODIS) onboard the Aqua satellite. One is the CALIOP retrieval, which estimates the aerosol composition from the CALIOP observation in the day and night time."* Have you upgrade the raw CALIOP retrievals or you just processing them for the comparison against those given by the CALIOP-MODIS retrieval?
7. **Line 79:** Why only these aerosol components are considered in the CALIOP-MODIS retrieval?
8. **Lines 96-97:** Please rephrase this sentence.
9. **Lines 96-102:** How do you match the VFM product and the regridded L1B CALIOP data since they are not reported at a common horizontal/vertical resolution? Have you performed an analysis showing how much "sensitive" are your results depending on the selected CAD score?
10. **Lines 120-121:** Specify the source of the Aeronet observations (sun-direct, almucantar retrievals, spectral deconvolution algorithm) that you are processing.
11. **Lines 163-165:** Can you elaborate further this statement?

12. **Lines 195-199:** This paragraph is not so clear and it is needed a rephrasing and a better explanation.

13. **Equation 17:** Do you mean temperature differences instead of variations of the heating rates?

14. **Lines 353-354:** The CALIOP-MODIS retrieval products are compared against CALIOP/MODIS observations? It is not clear to term "simulations" used here.

15. **Figure 5:** In the manuscript there is a description of what it is shown in scatterplots and interpretation is rather poor (Lines 385-388). Why the AOD532nm collocated data are clustered in vertical lines and they are not scattered as for AOD at 1024nm?

16. **Figure 7:** Can you provide an interpretation of the obtained findings?

17. **Figure 8:** It would be useful to discuss how the joint CALIOP-MODIS retrieval modify the raw AODs given by CALIOP and MODIS. For instance, the maximum AODs in the Bodele Depression reproduced by MODIS are not evident neither in the CALIOP patterns nor in the CALIOP-MODIS AODs. Is there any explanation on that? Moreover, in East Asia, the CALIOP-MODIS AODs are close to those given by MODIS. In general, it seems that the utilization of the MODIS data causes a convergence between the CALIOP and the CALIOP-MODIS AOD retrievals. For the CALIOP retrievals you are using the official products in which specific lidar ratios are implemented. However, these values might not be representative as it has been shown in Floutsi et al. (2023). Can you reproduce the CALIOP plots after implementing the upgraded lidar ratios?

18. **Figures 9-12:** The discussion in the manuscript focuses on the figures description without an interpretation of the key findings.

19. **Lines 451-456:** It is not clear which Aeronet data are used exactly. For the AOD is better to use the sun-direct measurements whereas for the other properties (SSA, AF, fine/coarse radii) you are relying on the almucantar retrievals. Is this correct? I think that the number of the collocated samples is very low (particularly for AOD). How many Aeronet stations are used? Can you provide a map depicting the Aeronet sites?

20. **Lines 514-518:** I think that it is missing here a comparison with other relevant studies (e.g., Korras-Carraca et al. (2022))

21. **Figure 17:** I am impressed with the predominance of the positive TOA DREs induced by LA particles over continents (in most parts) and in the outflow regions in the Tropical and the Southern Atlantic Ocean. Is this possible attributed to the low SSAs?

22. **Summary and conclusions:** I suggest to reduce the length of the text.

---

## Author Comment (AC1)

We thank you for taking the time to review the manuscript and for your helpful comments. We have revised the manuscript in response to your comments. We believe that the manuscript has been greatly improved thanks to your suggestions.

This is a very interesting paper utilizing state of the art satellite data from active and passive remote sensing sensors together with models in order to characterize aerosol optical properties globally, by clustering to aerosol types and relevant aerosol properties.
The novelty and the strength of the paper is the use of the less uncertain inputs from both satellite sensors and combine it in a model run, globally.

Introduction

I miss the state of the art on aerosol global climatologies based on Kinne et al and other AEROCOM related publications.Also, Amiridis et al., for calipso.   In addition, publications discussing dust only ODs based on MODIS-CALIPSO-MERRA2 synergies have been presented by Gkikas et al..

We added the following description of aerosol data sets in Sect. 1.
"Based on the recent sophisticated numerical models with aerosol modules, and space- and ground-based observations, the data sets of aerosol composition climatology have been developed. The Modern-Era Retrospective analysis for Research and Applications version 2 (MERRA-2; Gelaro et al., 2019), and the Copernicus Atmosphere Monitoring Service Reanalysis (CAMSRA; Innes et al., 2019), and the Japanese Reanalysis for Aerosol v1.0 (JRAero; Yumimoto et al., 2017) are the reanalysis data sets by data assimilation schemes. The Max-Planck-Aerosol Climatology version 2 (MACv2; Kinne et al., 2019) is a climatology data set created by merging the data of the Aerosol Robotics Network (AERONET; Holben et al., 1998) and MAN (Smirnov et al., 2009) ground-based sun-photometer networks onto the ensemble mean of AeroCom models (Kinne et al., 2006). These data sets provide the global distributions of major aerosols, such as, sulfate, organic carbon, BC, dust, and sea-salt. The ModIs Dust AeroSol (MIDAS; Gkikas et al., 2021) data set is the global map of dust at fine resolution (0.1º × 0.1º), and is created by the aerosol optical depth derived from the Moderate Resolution Imaging Spectroradiometer (MODIS) and the dust fraction of the MERRA-2 reanalysis. Amiridis et al. (2015) develop LIVA (Lidar climatology of Vertical Aerosol Structure for space-based lidar simulation studies), which is a three-dimensional multi-wavelength global aerosol and cloud optical data set. This data set is based on the Cloud-Aerosol Lidar with Orthogonal Polarization (CALIOP) on board the Cloud Aerosol Lidar Infrared Pathfinder Satellite Observations (CALIPSO) satellite (Winker et al., 2010), and the ground-based networks of European Aerosol Research Lidar Network (EARLINET; Bösenberg et al., 2003; Pappalardo et al., 2014) and AERONET."

In addition, the distinction of the aerosol types with mixing possibilities in the atmospheric is a basic assumption of the paper and it have to be accompanied by studies elaborating on different approaches of

In Sect. 1, we described the assumption of aerosol compositions in the remote sensing as follows: "In the previous remote sensing methods of aerosol compositions, there are two approaches in assuming aerosol components. One is the CALIOP-type categorization, such as, clean marine, polluted continental, and smoke, etc. These types are based on the aerosol characteristics observed in the typical scenes. The other is the similar categorization to the numerical models, i.e., sulfate, organic carbon, BC, DS, and SS. We adopted the latter approach because the external mixing of these components is applicable to various scenes, and the $\tau_a$ and extinction coefficient ($\alpha_a$) of each component are suited for the comparison with the numerical models and the data assimilation. In this study, aerosols are assumed to consist of four components with different sizes, light-absorbing features, particle mixtures, and shapes. We defined these components as WS, light-absorbing particles (LA), DS, and SS. WS is defined by an external mixture of sulfate, and organic carbon, etc., because both the sulfate and organic carbon are fine and less light-absorbing particles, and it is difficult to estimate sulfate and organic carbon separately from the MODIS and CALIOP measurements. LA is defined by an internal mixture of WS and BC. The details of the assumed aerosols are described in the Sect. 3. In this study, the global three-dimensional distributions of these components were estimated from the CALIOP-MODIS retrieval."

Schematic of the retrieval.

Looking at the figure 1 scheme. I was wondering how the optimized x step is achieved, only for part of the aerosol properties or satellite based observations used for the matching at the convergence stage. Or some more clarity needed on the paragraph lines 135 to 144.

We found some mistakes in Fig. 1 and corrected them. The logarithmic transformation is applied to x and y and the objective function is minimized by the iteration in ln(x) space. Therefore, Equation 1 is modified. The convergence criterion is described in the revised manuscript.

Section 3.1.3 It is necessary to introduce a number of assumptions here, so the authors to my opinion have done a good work. A discussion on overall uncertainties of the method would be nice for the reader. Realistically these retrievals and assumptions work much better in different parts of the world and worse in others based on the aerosol field complexity. Could the authors comment on such aspects ?
For example standard deviations in figure 6 I presume, is a mix of "easier" retrievals spatially and more difficult ones that cause these standard deviations. Final effect will be lonked with more uncertain retrievals in some areas and less in others.

It is difficult to comment on that, but the comparison with the aerosol data set of Kinne (2019) provided information on the points. In Sect. 5.1 of the revised manuscript, we compared the global distribution of

SSA with the global map of SSA in Kinne (2019) in the revised manuscript. The comparisons of SSA showed good agreements of SSA in the central and southern parts of South America, and the southern part of Africa. These are famous biomass-burning regions. We think the assumptions regarding to LA would work better in these regions. However, the SSA was underestimated in the most parts of the land area.

We are now conducting the intensive validation study using the ground-based networks of AERONET, SKYNET (sun/sky photometer), and AD-Net (lidar) data. The validation study will show that our assumptions work better in different parts of the world and worse in others.

Figure 8. MODIS standard AOD have been used in various studies and has been extensively validated with AERONET data. (Moreover, MODIS itself use AERONET to retrieve (some kind of) uncertainty estimation over land and ocean). What is the novelty here with the use of Callipso in AOD only ? Is lower than MODIS standard global AOD more realistic ? And what improvements and errors are dealt here with the combined MODIS-Callipso retreival ?

The merged AOD of the dark target and deep blue algorithm in the MODIS collection 6.1 is slightly greater than the AERONET AOD (Shi et al., 2019). The AOD of the CALIOP has a negative bias (Kim et al., 2018). Our retrieved AOD exists between the CALIOP and MODIS standard products, and is better than the CALIPSO products. However, the bias of the CALIPSO and MODSI standard products is different by regions. We need further comparisons in the regional scale. We added these discussions in Sect. 5.1.

Figure 9 shows a very limited spatial varability of both parameters. First of all should be nice to increase the size and improve the quality of this figure as details can be already there but not visible.

The color bars of Figure 9 have been corrected to emphasize the spatial variations.

In general it would be nice to comment on difference the authors find compared with the Kinne aerosol climatologies. Discussion could be combined with figure 13 results.

In Sect. 5.1, we added the comparisons of SSA, AF, and compositions with the global maps of MACv2 (Kinne, 2019). The global distributions of SSA and AF matched well to MACv2, but the SSA over the most parts of the land area is smaller than that of MACv2. The underestimation of SSA is also seen in the comparisons with the AERONET (Fig. 13).

Figure 10 is the paper highlight and it needs technical improvements in order to be able to see spatial details and changes of the AODs.

The color bars of Figure 10 have been modified to emphasize the spatial variations.

Figure 14 is a very nice demonstration of aerosol shape effects.

Thank you very much.

I would see fig 14 to 18 in a supplement. But it is up to the authors to decide.

Figures 17 to 19 are important in this study, so only Figures 14 to 16 were moved to the supplement.

Fig. 19 should be discussed much more as properties like SSA are ery uncertain based on the figures 9 and 13.

In the revised manuscript, we discussed the overestimation of LA and its impact on the heating rate.

Major comment
How this method improve compared with other existing ones and what are the advantages that lead to it?

CALIOP provides the global 3-D distribution of aerosols, but the AOD is underestimated due to the low signal to noise ratio in the tenuous layers (Omar et al., 2013; Kim et al., 2018). The AOD of the MODIS has small positive bias (Shi et al., 2019). The synergistic retrieval using both CALIOP and MODIS observations provides the AOD close to the MODIS products, and the better global 3-D distribution of aerosols. In addition, our assumed aerosol components of WS, LA, DS, and SS are similar to the components defined in the numerical models with aerosol modules. The AOD and extinction coefficients of each component would be useful for the comparison with the numerical models, and the data assimilation. We added these descriptions in Sect. 1 and 6.

---

## Author Comment (AC2)

We thank you for taking the time to review the manuscript and for your helpful comments. We have revised the manuscript in response to your comments. We believe that the manuscript has been greatly improved thanks to your suggestions.

A nice structured work, developed by the utilization of CALIOP and MODIS retrievals for the establishment of a global aerosol-speciated 3D distribution. Typical aerosol properties are derived and collocated against ground-based stations (AERONET). Finally, SDRF values (under clear sky conditions) are retrieved and compared against results in previous studies for the estimation of aerosol induced perturbations on the Earth-Atmosphere radiation budget.

1 Introduction

I think the revised V4 types of CALIPSO and some weaknesses of CALIOP and MODIS retrievals - not only the limited wavelength information and the strong surface reflectance, respectively - should be mentioned (these preferences would probably have a reasonable contribution to the uncertainty in some CALIOP-MODIS retrievals).

We revised the sentences as follows: "The columnar properties of aerosols are available from the MODIS multi-wavelength information, and $\tau_a$ is retrieved accurately (e.g., Shi et al., 2019), but aerosol vertical profiles cannot be obtained, and strong surface reflection (e.g., snow, desert) makes the retrieval difficult (Hsu et al., 2013). CALIOP observations exclude the data at the layers contaminated by the surface reflection and provide information on the vertical profiles of aerosol optical properties and particle shapes (spherical/non-spherical), but only limited wavelength information. Additionally, CALIOP does not detect the tenuous layers in the daytime due to the low signal to noise ratio. This results in the underestimation of $\tau_a$ (Omar et al., 2013; Kim et al., 2018). The synergistic use of both instruments decreases the influences of the surface reflection and provide the more accurate columnar properties and vertical profiles of aerosols. Furthermore, the particle size information is obtained from the combined spectral information of the CALIOP and MODIS observations (Kaufman et al., 2003)."

5 Retrieval results from the CALIOP and MODIS observations in 2010

In Figure 8 the different strong aerosol sources (e.g. dust source in the region of Bodélé) are not visible. For example, a well-known problem of CALIOP-CALIPSO retrievals is the sufficient underestimation of AOD over strong aerosol sources, an inadequacy strongly related to the presence opaque layers completely attenuating the laser beam. Probably a colorbar with a lower AOD limit (less than 0.8) or with modified bins or just a different colorbar could help with the visualization of this result. If a filter is applied for the smoothness of the colors on the map, this filter maybe contaminates the AOD over the sources especially if the surrounding regions have substantially lower AOD.

Thank you for your advice. The color bars of Fig. 8 have been modified to emphasize the spatial variations. The dust source of Bodélé was clear in the MODIS result (Fig. 8c). Although the CALIOP-MODIS retrieval utilizes the MODIS measurements, the dust source was not clear in the CALIOP-MODIS and CALIOP results (Figs 8a and b). We think the sparse observation of the CALIOP in the longitude direction may be a possible cause. This discussion was added in the revised manuscript.

In Figure 9 an aerosol-speciated distribution is not clear. It's like having 2 groups of SSA values (land-ocean). A narrower colorbar (starting e.g. from 0.8) could help with the distinguishing of some areas. For example, over the Northern and the Central Africa a lower and a higher SSA value should be visible (dust and more absorbing particles-like smoke from biomass burning- respectively). The same problem is visible for AF.

The color bars of Figure 9 have been modified, and the spatial variations were discussed in the revised manuscript as follows: "Figure 9 shows the horizontal distributions of $\omega_0$ and $g$ of the CALIOP-MODIS retrieval. The global means of $\omega_0$ and $g$ were about 0.940 $\pm$ 0.038 and 0.718 $\pm$ 0.037. Previous studies have shown that the global mean $\omega_0$ is from 0.89 to 0.953 (Korras-Carraca et al., 2019; Kinne, 2019), and the global mean $g$ is 0.702 (Kinne, 2019). Our results are thus consistent with these previous studies. $\omega_0$ over the land was from 0.8 to 0.95 and was smaller than that over the ocean. $g$ over the land was from 0.6 to 0.75 and also smaller than that over the ocean. These differences between land and ocean are due to the presence of SS over the ocean, because $\omega_0$ and $g$ of SS are larger than those of the other aerosol components (Table 1). In the major biomass-burning regions of the central and southern parts of South America, and the southern part of Africa, $\omega_0$ and $g$ of the CALIOP-MODIS retrieval are particularly small, from 0.85 to 0.90, and 0.65 to 0.70, respectively. These are consistent with the results of Kinne (2019). However, our retrieved $\omega_0$ is less than 0.90 over the most parts of the land area and appears to be about 0.05 smaller than $\omega_0$ of Kinne (2019). In Sect. 4, it was shown that the CALIOP-MODIS retrieval tended to underestimate $\omega_0$. The tendency to underestimate $\omega_0$ might appear in the retrieval over the land."

In Figure 12 it's not clear for me some hotspots of coarse DS particles over the Norway and Sweden

The color bars of Figure 12 have been modified. There are two hotspots over the Norway and Sweden. However, there are no major desert region in Norway and Sweden, and the dust AOD is small in Fig. 10c. The uncertainties of the retrievals become large in the small AOD cases, and the particle radius of DS tends to be overestimated (Fig. 6l). The particle radius of DS would be overestimated at the hotspots. There are many hotspots in Fig. 12c, and the retrieved particle radius of DS deviates significantly from the AERONET data (Fig. 13f). We need the further investigations of validation and quality control.

In Figure 13 AOD shows a good agreement with AERONET, but the other parameters rather deviate. In

comparison with Figure 9 maybe the results for the other properties need further investigation, since these parameters are also used for the radiative simulations and furthermore for the heating rate.

In this study, we used only the data in 2010. We are now processing the data from 2007 to 2021, and we will conduct the intensive validation study using the ground-based networks of AERONET, SKYNET (sun/sky photometer), and AD-Net (lidar). We will improve the constraints and assumptions in the CALIOP-MODIS retrieval after the validation study.

---

## Author Comment (AC3)

We thank you for taking the time to review the manuscript and for your helpful comments. We have revised the manuscript in response to your comments. We believe that the manuscript has been greatly improved thanks to your suggestions.

AGeneral comment

This is an interesting paper which develops an optimal estimation aerosol retrieval using combined CALIOP and MODIS observations. The retrieval attempts to retrieve effective aerosol size and optical properties for each of four aerosol types. This goes far beyond the standard retrievals of CALIOP and MODIS, or any other satellite sensor, so it is good that comparisons with Aeronet retrievals are included to evaluate the performance of the retrieval. The algorithm is described well and the authors do a good job of examining some of the uncertainties (particle model assumptions) but more details on uncertainties (Section 4) and a few other topics would be helpful.

Specific comments

CALIOP Level 1B data is pre-processed by calculating running means using horizontal averaging over 10 km. Are retrievals performed only on 10-km averages that do not contain clouds, or are cloudy profiles removed before averaging to 10 km?

The running means were performed after the cloud layer data was removed by the VFM. We have revised Sect. 2.1.

The authors mention several times in Section 3 that the DVCs and DMRS are 'optimized to all CALIOP and MODIS measurements'. It is not clear to me what this means. Are the retrieved parameters adjusted to minimize the merit function for each MODIS-CALIOP data pair, or is there some sort of global optimization which is performed?

"each MODIS-CALIOP data pair" is correct. We corrected them in the manuscript.

What is the altitude range of the CALIOP-MODIS retrieval? It appears to be 0-10 km from Figure 8b.

The upper limit of the altitude is about 30 km. This is described in Sect. 2.1 of the revised manuscript.

I'm not sure how to read Table 3. What is meant by "relative value"? Relative to what? In the first row, is the mean difference between retrieved and simulated values an AOD of -0.15 or is it 15% of something?

The equations for the calculations of the differences were added to Tables 3 and 4, and the relative values

were expressed as percent values in the revised manuscript.

Lines 284-285: How does it follow that the AOD of WS is greater than the AOD of LA because the SSA of LA is lower?  The scattering extinction coefficient is reduced when SSA decreases, but the total extinction is unchanged.  Please explain.

We added the following description in Sect. 3.1.3.
"In the AERONET product at worldwide locations, $\omega_0$ ranges from 0.8 to 1.0 (Dubovik et al., 2002). $\omega_0$ is about 0.96 for WS and about 0.44 for LA (Table 1), and $\omega_0$ for an external mixture of WS and LA is calculated by $\omega_0 = (\tau_{a,WS}\omega_{0,WS} + \tau_{a,LA}\omega_{0,LA})/(\tau_{a,WS} + \tau_{a,LA})$. Thus, $\tau_a$ of WS must be greater than that of LA."

To understand the realism of the retrieval simulations more detail should be added on how the satellite Level 1 data was simulated in Section 4.  From Line 256, it appears that a single number is used for random error in lidar backscatter (15%).  But the relative random error of CALIOP attenuated backscatter profiles varies with altitude and with the albedo of the underlying surface and can be much worse than 15%, especially at higher altitudes.  Was noise from the solar background simulated and added as a random variable to each sample in the vertical profile?  Were retrieval errors due to systematic MODIS calibration errors or estimates of surface albedo considered?

The altitude dependency, surface contribution, and solar background noises should be included in the simulations. However, the information available from the published papers is limited, particularly, for the CALIPSO version 4 data set. It is difficult to simulate these realistic noises. The measurement accuracy of the attenuated backscatter coefficient at 532 nm in the version 3 product is evaluated by the comparison with the airborne HSRL. The mean difference is 2.9 % and the standard deviation is 20 % (Rogers et al., 2011). The bias of the version 4 product is smaller that the version 3 (Getzewich et al., 2018). Furthermore, we smoothed the data by the running mean, and the CALIOP-MODIS retrieval optimizes the state vector to only the data discriminated as aerosols by the VFM. Therefore, we think the random error of 15 % in the backscatter coefficient is an appropriate value.

The systematic errors of MODIS calibration and surface albedo were not considered. However, the random errors are important in this study. We use two bands of visible and near-infrared wavelengths. The volumes of each fine and coarse modes are estimated separately from the two bands. And the median radii of the fine and coarse modes depend on the spectral dependency of two bands. The random errors affect the spectral dependency of two bands, and make it difficult to retrieve the volumes and median radii of the fine and coarse modes.

We are now processing the observation data in the period from 2007 to 2021. We will conduct the further validation study using the ground-based remote sensing networks of AERONET, SKYNET, and AD-Net to estimate the practical uncertainties of the retrievals. We think the retrieval simulations in this study is

useful to interpret the retrieval errors in the validation study.

It is odd that retrieval uncertainties are larger over ocean than over land, while CALIOP and MODIS retrievals are both better over ocean than land. Is this really because SS is retrieved over ocean but not land, as the authors say, or could it be because AOD over ocean tends to be much smaller than over land? Or is it due to uncertainties in the optical model used for SS? It would be good to discuss reasons for this behavior in more detail. Marine aerosol is not just 'sea salt' and often contains internally mixed biogenic sulfate or biogenic organic compounds. This might impact the refractive index of the particle model used.

The simulated AOD is from 0.05 to 1.0 for both the land and ocean cases. The same optical model of SS is used in the simulations and retrievals. However, the particle radius of SS is different in the simulation and retrieval because the particle radius of SS is determined by the ocean surface wind speed and the random errors are added to the ocean surface wind speed in the simulations. The particle radius of SS is not optimized in the retrieval. Therefore, the difference of the particle radius of SS affects the AOD of SS. Since both WS and SS are less light-absorbing particles, the AOD of WS is overestimated (underestimated) when the AOD of SS is underestimated (overestimated). This different sign of the retrieved AOD can be seen in Table 3. We added these descriptions in Sect. 4.2.
WS would be estimated together with SS, if the biogenic sulfate or organic compounds have similar optical properties to WS. Actually, WS is estimated over the ocean, and the distribution of WS over the ocean is similar to the distribution of SS (Fig. 10a). WS over the ocean may be the biogenic sulfate and organic compounds, and fine particle of sea salt.

The authors comment that extinction coefficients are unnaturally large at 70N and 70S-80S. I do not see evidence of this in figure 8a or 8b and am wondering what the authors are referring to. There appear to be very few retrievals at 70S-80S. The text says the large EC are due to cloud contamination, but could it be due to ice cover and thus high surface albedo? Are retrievals attempted over ice or only over ice-free ocean?

The color bar of Fig. 8 was modified to emphasize the spatial variations. The slightly large ECs are seen at the altitudes from 0 to 9 km and latitudes from 70S to 80S, and are indicated by blue color in Fig. 8b of the revised manuscript. A peak of EC is seen at altitudes from 0 to 1 km and latitudes at 70N, and are indicated by the colors from green to red. The retrievals have been attempted to the observations over the ice surface. As you say, it is possible that the high surface albedo of ice has affected the retrieval of the ECs. We mentioned the influences of high surface albedo of ice in the revised manuscript.

Minor comments

It was not clear to me what is meant by 'dry volume concentration' (line 125). What are the units?

The unit is um$^3$/um$^3$. We added the following explanation in Sect. 3.1.1.

"$V_{dry}$ is defined as the volume of aerosols at a relative humidity of 0 % per unit atmospheric volume, and $r_{m,dry}$ is defined as the median radius of aerosols at a relative humidity of 0 %."

Equations 10 and 13 explain constraints applied to the solution, using somewhat different approaches to notation. I find the approach used in Eqn 13 to be more clear than Eqn 10.

Eqs. 10 and 11 are changed to the same approach as Eq. 13.

Lines 422-423: rather than "SSA of the land ..", I think "SSA over land .." is meant, and the same for "of the ocean" and for AF

We corrected them.

The authors introduce a large number of non-standard 2- and 3-letter abbreviations for various parameters (ABC, LR, DMR, ⋯) , and then later introduce mathematical symbols for some of these parameters when used in equations. It would be simpler to define the math symbols and use them throughout the paper. I found DMR and DVC especially awkward and had a much easier time reading re and vdry .

We changed the abbreviations to the mathematical symbols.

Depolarization ratio (DR) and linear depolarization ratio (LDR) are both used. Aren't these the same parameter?

We have used DR for the CALIOP measurements, and LDR for particle optical property, but these are confusing to readers. In the revised manuscript, the total depolarization ratio ($\delta$) was used for the CALIOP measurements, and the particle depolarization ratio ($\delta_a$) was used for the particle optical property.

---

## Author Comment (AC4)

We thank you for taking the time to review the manuscript and for your helpful comments. We have revised the manuscript in response to your comments. We believe that the manuscript has been greatly improved thanks to your suggestions.

The current study deals with the development of a joint retrieval algorithm utilizing CALIOP and MODIS observations. In the algorithm, four aerosol species (water-soluble, light-absorbing, dust and sea-salt) are assumed and for each one of them optical and microphysical properties are derived for 2010. The obtained products are compared against those provided by the official CALIOP and MODIS products as well as against AERONET observations. Moreover, the authors have performed an analysis of the aerosol-induced direct radiative effects. Overall, it is an interesting study which has a lot of potential. However, there are several parts in the manuscript which must be improved thus helping the reader to understand the approach and the scope of each step of the applied methodology (Section 3.1). Another weak point of the study is the poor interpretation of the results presented in Section 5. To be more specific, the discussion should not be limited only in the description of the plots but it should be associated with a physical interpretation. I would also suggest the authors to revise the English writing style throughout the text. As a conclusion, I believe that the submitted paper fits to the AMT purposes but it is needed a major revision before being accepted for publication. I hope that my comments, which are listed below, will help the authors to improve their work.

1. Line 35: Aerosols interact also with the longwave radiation.

We changed "solar radiation" to "solar and terrestrial radiation".

2. It is better to use the term radiative effect rather than radiative forcing since the latter one is related to the induced perturbations of the radiation budget attributed to anthropogenic activities.

We changed "SDRF" to "SDRE".

3. Lines 46-48: Since you are mentioning the Aeronet optical properties derived by almucantar retrievals you must cite the relevant publications from Oleg Dubovik.

We added the reference papers of Dubovik and King (2000), Dubovik et al. (2006), and Synuk et al. (2020).

4. Lines 51-53: Can you be more specific? Which imagers?

SeaWIFS, MODIS, and OMI are used in the retrieval methods of Higurashi and Nakajima (2002) and Kin et al. (2007).

5. Lines 55-58: Why is important to mention the aerosol subtypes in the CALIOP Version 3 data since they are not used in the current study?

The description of the version 3 was removed.

6. Lines 70-81: I suggest to describe explicitly which are the advantages and disadvantages of the CALIOP and MODIS instruments and how complement each other. The current description is poor and vague. It is needed a better description supported by findings of previous studies. What do you mean "To observe the global three-dimensional distribution of the aerosol composition, we have developed two aerosol composition retrieval methods that use the observations of CALIOP and the Moderate Resolution Imaging Spectroradiometer (MODIS) onboard the Aqua satellite. One is the CALIOP retrieval, which estimates the aerosol composition from the CALIOP observation in the day and night time." Have you upgrade the raw CALIOP retrievals or you just processing them for the comparison against those given by the CALIOP-MODIS retrieval?

We developed the CALIOP retrieval based on the similar algorithm to the CALIOP-MODIS retrieval, and the method is different from the raw CALIOP retrievals. We are now preparing the paper of the CALIOP retrieval. However, the description of the CALIOP retrieval is confusing to the reader. We removed the description in the revised manuscript.

Following your comments, we have modified the paragraph as follows: "To observe the global three-dimensional distribution of the aerosol composition, we have developed a new aerosol composition retrieval method that use the CALIOP and MODIS observations. The CALIOP-MODIS retrieval optimizes the aerosol composition to both the CALIOP and MODIS observations in the daytime. The columnar properties of aerosols are available from the MODIS multi-wavelength information, and $\tau_a$ is retrieved accurately (e.g., Shi et al., 2019), but aerosol vertical profiles cannot be obtained, and strong surface reflection (e.g., snow, desert) makes the retrieval difficult (Hsu et al., 2013). CALIOP observations exclude the data at the layers contaminated by the surface reflection and provide information on the vertical profiles of aerosol optical properties and particle shapes (spherical/non-spherical), but only limited wavelength information. Additionally, CALIOP does not detect the tenuous layers in the daytime due to the low signal to noise ratio. This results in the underestimation of $\tau_a$ (Omar et al., 2013; Kim et al., 2018). The synergistic use of both instruments decreases the influences of the surface reflection and provide the more accurate columnar properties and vertical profiles of aerosols. Furthermore, the particle size information is obtained from the combined spectral information of the CALIOP and MODIS observations (Kaufman et al., 2003)."

7. Line 79: Why only these aerosol components are considered in the CALIOP-MODIS retrieval?

Sulfate, black carbon, organic carbon, dust, and sea salt are assumed in most numerical models. We

introduced the similar components to the numerical models for the comparison with the numerical model results, and the data assimilation in the future. It is difficult to retrieve sulfate and organic carbon separately, because sulfate and organic carbon are fine and less light-absorbing particles. Therefore, water-soluble is defined as externally mixed aerosols of sulfate, organic carbon, etc. This is described in Sect. 1.

8. Lines 96-97: Please rephrase this sentence.

We have revised the paragraph.

9. Lines 96-102: How do you match the VFM product and the regridded L1B CALIOP data since they are not reported at a common horizontal/vertical resolution? Have you performed an analysis showing how much "sensitive" are your results depending on the selected CAD score?

Firstly, the cloud contaminated data of the L1B CALIOP data were removed by the VFM. Then, the L1B data within the horizontal and vertical windows of 10 km and 120 m were collected and averaged. We revised Sect. 2.1.
At the beginning of this study, the CAD score of 20 was used to discriminate clouds and aerosols. However, many hotspots of ECs were found in the retrieval results. Therefore, a more rigorous score of 70 was adopted.

10. Lines 120-121: Specify the source of the Aeronet observations (sun-direct, almucantar retrievals, spectral deconvolution algorithm) that you are processing.

We used the almucantar retrievals. The source was specified in the revised manuscript.

11. Lines 163-165: Can you elaborate further this statement?

In the revised manuscript, we described the aging processes of BC particles at the beginning of the paragraph, and added the reference of the observed spatiotemporal changes of BC mixing state.

12. Lines 195-199: This paragraph is not so clear and it is needed a rephrasing and a better explanation.

The paragraph was modified as follows: "To reduce the computational time, we constructed the lookup tables of $\alpha_a$, $\omega_0$, and the phase matrix for each model using the above-mentioned particle models and size distributions. The inputs of the lookup tables were $V_{dry}$ and $r_{m,dry}$ of WS, LA, DS, and SS, and relative humidity. The outputs were $\alpha_a$, $\omega_0$, the phase matrix, and the size distribution of each component at the input relative humidity. Finally, $\alpha_a$, $\omega_0$, phase matrix, $g$, $S_p$, $\delta_p$, and size distribution of total aerosols (WS+LA+DS+SS) were calculated according to the external mixture. These optical properties are used in

the forward models of CALIOP and MODIS observations.

13. Equation 17: Do you mean temperature differences instead of variations of the heating rates?

We calculated the difference of the heating rate with and without aerosols. "T" in the equation was replaced by "HR" in the equation.

14. Lines 353-354: The CALIOP-MODIS retrieval products are compared against CALIOP/MODIS observations? It is not clear to term "simulations" used here.

We applied the CALIOP-MODIS retrieval to the synthetic data of the CALIOP and MODIS observations, which was created by the simulations using the forward models in Sect. 3. We modified the sentences.

15. Figure 5: In the manuscript there is a description of what it is shown in scatterplots and interpretation is rather poor (Lines 385-388). Why the AOD532nm collocated data are clustered in vertical lines and they are not scattered as for AOD at 1024nm?

The plots of AOD at 532 nm are aligned vertically in the lines because we controlled the total volume of aerosols by giving AOD at 532 nm in the simulations.
The detailed explanation for the retrievals over the ocean was added in the revised manuscript as follows: "In general, the small value of the ocean surface albedo is an ideal situation for the satellite remote sensing of aerosols. However, the retrieval results for $\tau_a$ of WS over the ocean are worse than those over the land because SS is taken into account, in addition to WS, LA, and DS, in the ocean surface cases. In the simulations, the random errors are added to the ocean surface wind speed. Since $r_{m,dry}$ of SS is determined by the given ocean surface wind speed and is not optimized in the CALIOP-MODIS retrieval, the random errors cause the difference of $r_{m,dry}$ of SS between the simulation and retrieval. The difference affects $\tau_a$ of SS. Since both WS and SS are less light-absorbing particles, $\tau_a$ of WS is overestimated (underestimated) when $\tau_a$ of SS is underestimated (overestimated). This opposite sign is seen in the ocean cases of Table 3."

16. Figure 7: Can you provide an interpretation of the obtained findings?

The EC of LA and SS were overestimated and the EC of WS and DS were underestimated at all altitudes. The overestimations of LA and SS were compensated by the underestimation of WS and DS. Consequently, the errors for the EC of total aerosols were small (Table 4.). We described these in the revised manuscript.

17. Figure 8: It would be useful to discuss how the joint CALIOP-MODIS retrieval modify the raw AODs given by CALIOP and MODIS. For instance, the maximum AODs in the Bodele Depression reproduced

by MODIS are not evident neither in the CALIOP patterns nor in the CALIOP-MODIS AODs. Is there any explanation on that? Moreover, in East Asia, the CALIOPMODIS AODs are close to those given by MODIS. In general, it seems that the utilization of the MODIS data causes a convergence between the CALIOP and the CALIOP-MODIS AOD retrievals. For the CALIOP retrievals you are using the official products in which specific lidar ratios are implemented. However, these values might not be representative as it has been shown in Floutsi et al. (2023). Can you reproduce the CALIOP plots after implementing the upgraded lidar ratios?

We described the regional distributions of AOD in the revised manuscript. In the global scale, the CALIOP-MODIS retrieval was between the CALIOP and MODIS standard products, but the results were different by the regions.

The dust source of the Bedele depression in the Sahara was not clear in the CALIOP standard and CALIIO-MODIS retrieval products. The CALIOP-MODIS retrieval utilizes the MODIS measurements but the dust source of the Bedele depression was not clear. We think this may be due to the sparse observations of the CALIOP in the longitude direction.

Do you mean the retrieval of the extinction coefficient from the CALIOP L1B data using the lidar ratios of Floutsi et al. (2023)? It is not easy for us and is beyond the scope of this study.

18. Figures 9-12: The discussion in the manuscript focuses on the figures description without an interpretation of the key findings.

We discussed the results compared with the global maps of Kinne (2019), Gkikas et al. (2021), and Korras-Carraca et al. (2021) in the revised manuscript.

19. Lines 451-456: It is not clear which Aeronet data are used exactly. For the AOD is better to use the sun-direct measurements whereas for the other properties (SSA, AF, fine/coarse radii) you are relying on the almucantar retrievals. Is this correct? I think that the number of the collocated samples is very low (particularly for AOD). How many Aeronet stations are used? Can you provide a map depicting the Aeronet sites?

We used the AOD derived from the sun direct measurements, and the SSA, AF, fine/coarse radii of the almucantar retrievals in the Level 2 data product. The sampling number is 91 for the 51 AERONET stations. We added the map of the AERONET stations as a supplement (Figure S1). The CALIOP does not has swath observations, and our retrieval was conducted every 5 km along the track of the CALIPSO satellite. Therefore, the sampling number is low. We are now processing the CALIOP and MODIS data from 2007 to 2021. We will conduct the further validation study using the long-term data.

20. Lines 514-518: I think that it is missing here a comparison with other relevant studies (e.g., Korras-

Carraca et al. (2022))

We compared the SDRE of this study with the results of Kinne (2019) and Korras-Carraca (2021) and discussed the differences.

21. Figure 17: I am impressed with the predominance of the positive TOA DREs induced by LA particles over continents (in most parts) and in the outflow regions in the Tropical and the Southern Atlantic Ocean. Is this possible attributed to the low SSAs?

Yes. Korras-Carraca et al. (2021) also shows the similar distribution of the positive TOA DRE.

22. Summary and conclusions: I suggest to reduce the length of the text.

We reduced the length of the text.